



**Fate of rice shoot and root residues, rhizodeposits, and microbe-assimilated**
**carbon in paddy soil: I. Decomposition and priming effect**
**Zhenke Zhu[1,2,#], Guanjun Zeng[2,#], Tida Ge[1,2], Yajun Hu[1], Chengli Tong[1], Olga Shibistova[3,4], Juan**
**Wang[1], Georg Guggenberger[1,3], and Jinshui Wu[1,2]**
[1]Key Laboratory of Agro-ecological Processes in Subtropical Region, Institute of Subtropical
Agriculture, Chinese Academy of Sciences, Hunan, 410125, China
[2]Changsha Research Station for Agricultural and Environmental Monitoring, Institute of Subtropical
Agriculture, Chinese Academy of Sciences, Hunan, 410125, China
[3]Institute of Soil Science, Leibniz Universität Hannover, 30419 Hannover, Germany
[4]VN Sukachev Institute of Forest, Siberian Branch, Russian Academy of Science, 660036 Krasnoyarsk,
Russian Federation
# These authors contributed equally to this work.
*Correspondence to:* Tida Ge (gtd@isa.ac.cn) and Jinshui Wu (jswu@isa.ac.cn)





**Abstract.** The input of recently photosynthesized C has significant implications on soil organic
carbon sequestration, and in paddy soils, both plants and soil microbes contribute to the overall C input.
In the present study, we investigated the fate and priming effect of organic C from different sources by
conducting a 300-d incubation study with four different $^{13}$C-labelled substrates: rice shoots (Shoot-C),
rice roots (Root-C), rice rhizodeposits (Rhizo-C), and microbe-assimilated C (Micro-C). The efflux of
both $^{13}CO_2$ and $^{13}CH_4$ indicated that the mineralization of C in Shoot-C-, Root-C-, Rhizo-C-, and
Micro-C-treated soils rapidly increased at the beginning of the incubation and then decreased gradually
afterwards. In addition, the highest level of C mineralization was observed in Root-C-treated soil
(45.4%), followed by Shoot-C- (31.9%), Rhizo-C- (7.9%), and Micro-C-treated (7.7%) soils, which
corresponded with mean residence times of 33.4, 46.1, 62.9, and 192 d, respectively. Furthermore, the
cumulative mineralization of native soil organic carbon in Shoot-C-treated soils was 1.48- fold higher
than in untreated soils, and the priming effect of Shoot-C on $CO_2$ and $CH_4$ emission was strongly
positive over the entire incubation. However, Root-C failed to exhibit a significant priming effect,
which suggests that it could potentially be used to mitigate $CH_4$ emission. Although the total C contents
of Rhizo-C- (1.89%) and Micro-C-treated soils (1.9%) were higher than those of untreated soil (1.8%),
no significant differences in total C emissions were observed. However, the $^{13}$C emissions of Rhizo-C-
and Micro-C-treated soils gradually increased over the entire incubation period, which indicated that
soil organic C-derived emissions were lower in Rhizo-C- and Micro-C-treated soils than in untreated
soil, and that rhizodeposits and microbe-assimilated C could be used to reduce the mineralization of
native soil organic carbon and to effectively improve soil C sequestration. The contrasting behaviours
of the different photosynthesized C substrates suggests that recycling rice roots in paddies is more



beneficial than recycling shoots and reveals the importance of increasing rhizodeposits and
microbe-assimilated C in paddy soils *via* nutrient management.

**Keywords:** Paddy soil; Rice; Plant residues; Rhizodeposits; Microbe-assimilated carbon; $CO_2$ and
$CH_4$ emission; Priming effect





## 1    Introduction


The soils of rice paddies, which cover an area of ~165 million ha worldwide, hold great potential for
expanded C sequestration (Conrad et al., 2012; Ge et al., 2012; Lal, 2004), and the soil organic carbon
(SOC) pools in agricultural systems, of which plant C is the primary substrate, are significantly
affected by the input of crop residues (Weintraub et al., 2007). For example, after crops are harvested
or die, aboveground biomass, such as straw, stubble, and other surface debris, contribute to annual C
inputs (Lu et al., 2003), and photosynthesized C substrates are continuously released by rice plants as
rhizodeposits, such as decaying roots, throughout the growing season (Lu et al., 2002, 2003). However,
autotrophic soil microbes that assimilate $CO_2$ contribute to C sequestration in paddy soil, as well (Ge et
al., 2013; Yuan et al., 2012a), and since C inputs promote microbial activity and native SOC
decomposition (Ye et al., 2015) and the quantity and quality of such inputs influence microbe-mediated
decomposition processes (Brant et al., 2006; Creamer et al., 2015), the quantification of different C
substrates allocated to paddy soils and their respective effects on native SOC require further
investigation.

58        The aboveground biomass and root systems of rice plants represent the only inputs of available

organic C to paddy SOC (Johnson et al., 2006), the quantity and quality of which has been reported
previously (Chen et al., 2014; Kisselle et al., 2001; Zhang et al., 2015). However, although
aboveground biomass has been reported to make significant contributions to SOC sequestration (Lu et
al., 2003), rice roots have been reported to contribute 1.5–3-fold more C to SOC than shoots (Hooker et
al., 2005), and similarly, Molina et al. (2001) reported that the stalks and leaves of corn contribute 50%
less C to SOC than the roots and rhizodeposits. The predominant contribution of crop roots to SOC can
partly be explained by the chemical composition of roots, which includes cellulose and lignin, as well
as by residue–soil interactions, such as aggregate formation, which physically protect organic C from
biodegradation (Baumann et al., 2009; Johnson et al., 2006; Lu et al., 2003).

68        Previous studies have also reported that the rhizodeposits of rice account for ~17% of the

photo-assimilates (Nguyen, 2003) that enter paddy soil, and that rice rhizodeposits include soluble
exudates, root border cells, dead debris, and insoluble mucilage (Lu et al., 2003). In cereal crops,
10–25% of root exudates are incorporated into SOC, and rhizodeposits are thought to play a key role in
C cycling and sequestration in plant–soil–microbe systems (Kuzyakov, 2002; Kuzyakov et al., 2003).





In addition to the photosynthesized C substrates of plants, soil microbes are also able to assimilate $CO_2$
*via* the Calvin-Benson-Bassham cycle and, thus, can significantly contribute to the net uptake and
assimilation of atmospheric $CO_2$, as well (Ge et al., 2013; Yuan et al., 2012b). In fact, the $CO_2$ uptake
by phototrophic soil microbes has been reported to account for up to 0.36% of the total C fixed in rice
paddy soils and 0.19% of the total C fixed in upland soils (Ge et al., 2013; Yuan et al., 2012b).

78         However, the effect of C input from different C sources on the balance and stability of SOC has

received limited attention. For example, low-molecular-weight C substrates are protected from
mineralization *via* sorption onto soil particles (Jones and Edwards, 1998), which contributes to the
stability and sequestration of SOC, whereas the input of fresh organic C, such as green manure, straw,
and rhizodeposits, promotes the decomposition of native SOC and results in the emission of $CO_2$ and
$CH_4$ (Huo et al., 2013; Yuan et al., 2014c). In addition, different C substrates can also have stimulating
or restraining effects on the mineralization of native SOC, which are known as positive or negative
priming effects (PEs), respectively (Kuzyakov, 2010). Priming is often caused by the addition of
substrates with relatively high C availability and nutrient contents, which results in increased microbial
activity (Blagodatsky et al., 2010); however, such easily degraded compounds greatly enhance the
decomposition of native SOC (Blagodatsky et al., 2007; Qiao, et al., 2014), compared with the effects
of ryegrass, cellulose, or wheat straw, which have complex structures that are less available to microbes
(Kuzyakov and Bol, 2006; Kuzyakov et al., 2000).

91         Accordingly, the quantity and quality of different C inputs, as well as their fate and PE in paddy

soils, are globally important (Bastida et al., 2013; Johnson et al., 2006; Wang et al., 2015). Although
numerous studies have estimated the fate of plant residues and rhizodeposits in paddy soils, to our
knowledge, there is no comparative information on (1) the decomposition of different organic C
sources, such as rice shoots and roots, rhizodeposits, and microbe-assimilated C; or (2) the effects
different organic C sources on the mineralization of native SOC. We hypothesized that the
decomposition of such material would decrease with the increasing complexity of substrate
composition and that the PE of plant residues was stronger than that of rhizodeposits and
microbe-assimilated C, owing to their relatively higher quantity and stability in the soil. To investigate
these hypotheses by quantifying the contribution of different organic C sources to $CO_2$ and $CH_4$





emission and by analysing their PE, we conducted a 300-d incubation study using $^{13}$C-labelled rice
plant residues, rhizodeposits, and microbe-assimilated C in paddy soils.

**2    Materials and methods**
**2.1  Study site and soil sampling**
The experimental rice field was located at the Changsha Research Station for Agricultural and
Environmental Monitoring, Hunan, China (113°19′52″ E, 28°33′04″ N; 80 m above sea level), where
the soil was a typical stagnic anthrosol, developed from highly weathered granite, and the climate was
subtropical, with a mean annual temperature and rainfall of 17.5 ℃ and 1300 mm, respectively.
Moist soil samples were collected from the plough layer (0–20 cm) and sieved (<4 mm) to remove
visible plant residues. The soil contained 18.1 g kg$^{-1}$ organic C, 1.8 g kg$^{-1}$ total N, and 0.4 g kg$^{-1}$ total K
and had a pH of 5.6 and a soil: water ratio (w/v) of 1: 2.5.

**2.2  Production of $^{13}$C-labelled substrates**
Rice cultivation and $^{13}CO_2$ labelling were performed as described by Ge et al. (2012; 2013), with some
modifications. Briefly, 60 pots were filled with 1 kg dry soil, and of these, 40 pots were each planted
with three 30-d-old rice seedlings (*Oryza sativa* L. 'Zhongzao 39'), whereas the remaining 20 pots
were unplanted. Weeds were removed manually.
For $^{13}$C labelling, 30 pots (20 planted, 10 unplanted) were transferred to an automatically
controlled gas-tight growth chamber (110 cm length, 250 cm width, 180 cm height) and exposed to
$^{13}CO_2$-fumigation for 18 d (May 14–31, 2013), during the vegetative growth period (including the
entire tillering stage). The growth chambers were placed in a rice field to ensure that the environmental
conditions of the labelled and control plants would be identical for labelled plants and unlabelled
controls , and the remaining 30 pots (20 planted, 10 unplanted), which served as controls for measuring
natural $^{13}$C abundance, were placed 10–15 m from the growth chambers. The surface of each planted
pot was covered with black plastic sheeting, to prevent algal photosynthesis in the floodwater and to





ensure that only the rice shoots were exposed to $^{13}CO_2$ (i.e., not phototrophic microbes in the soil or
water), whereas the unplanted pots were left uncovered, so that the soils were directly exposed to $^{13}CO_2$
and so phototrophic soil microbes could assimilate atmospheric $^{13}CO_2$. All the pots were watered every
few days, in order to maintain a water depth of 2–3 cm above the soil surface, until harvest.
The $CO_2$ concentrations of the growth chambers were measured using an infrared analyser
(Shsen-QZD, Qingdao, China) and maintained at 360–380 μl L$^{-1}$. The $^{13}CO_2$ was generated by
acidifying $Na_2{}^{13}CO_3$ (1.0 M, 99 atom % $^{13}C$; Cambridge Isotope Laboratories, Tewksbury, MA, USA)
with $H_2SO_4$ (0.5 M) in beakers that were placed inside the growth chambers. During the labelling
period, $^{13}CO_2$ was only released when $CO_2$ concentrations fell below 360 μl L$^{-1}$, and at $CO_2$
concentrations >380 μl L$^{-1}$, the gas flow was diverted and passed through $CO_2$ traps (NaOH solution).
An air-conditioning system was used to control the temperature inside the chamber within 1 ℃ of the
ambient temperature in the rice field. Two fans continuously circulated the air in the growth chamber.

**2.3  $^{13}C$-labelled substrate collection**
All the rice plants and soils were sampled destructively after 18 d of $^{13}CO_2$ labelling. Rice shoots were
removed at their bases, whereas rice roots were separated from the soil by washing with deionized
water, and both shoots and roots were dried at 60 ℃ for 48 h and then cut into <5 mm pieces.
$^{13}C$-labelled rhizodeposits were obtained by gently shaking moist soil from the roots of rice plants and
were prepared for incubation by removing root debris and mixing thoroughly. Meanwhile, to obtain
microbe-assimilated $^{13}C$, we collected soil from $^{13}C$-treated, unplanted pots and mixed it thoroughly.

**2.4  Soil incubation**
To determine the PEs of different C sources and the effect of different C substrates on $CO_2$ and $CH_4$
emission, we conducted a 300-d incubation study of paddy soils that had been supplemented with
$^{13}C$-labelled shoots, roots, rhizodeposits, or microbe-assimilated C. Five treatments were used: (1)
unlabelled and unplanted paddy soil supplemented with $^{13}C$-labelled shoot residue (Shoot-C), (2)
unlabelled and unplanted paddy soil supplemented with $^{13}C$-labelled root residue (Root-C), (3)
$^{13}C$-labelled soil containing $^{13}C$-labelled rhizodeposits (Rhizo-C), (4) $^{13}C$-labelled soil containing
$^{13}C$-labelled microbe-assimilated C (Micro-C), and (5) unlabelled and unplanted soil without





supplementation (CK). Three additional treatments were used to determine the natural occurrence of
$^{13}C$: (1) unlabelled and unplanted paddy soil with unlabelled shoot residue, (2) unlabelled and
unplanted paddy soil with unlabelled root residue, and (3) unlabelled and unplanted paddy soil with
unlabelled rhizodeposits.
For the Shoot-C and Root-C treatments, 150 g (100 g dry weight equivalent) unlabelled,
unplanted soil with a water content of 50% was homogenized with 0.6 g of labelled and dried shoot and
root residue, respectively, with a final residue content of 6 g kg$^{-1}$. Subsequently, the samples were
transferred to 500 ml serum bottles with 100 ml deionized water, to ensure a water layer of >1–2 cm,
and the bottles were sealed with butyl rubber stoppers. For the Rhizo-C and Micro-C treatments, 150 g
fresh soil containing either $^{13}C$-labelled rhizodeposits (from rice roots) or $^{13}C$-labelled
microbe-assimilated C (from labelled, unplanted pots) were directly weighed into 500 ml serum bottles,
respectively. Incubation was conducted at 25 ℃ in the dark for 300 d, with four replicates for each
treatment. $CH_4$ and $CO_2$ concentrations of the headspace samples were collected at 1, 3, 5, 10 d and
then every 10 d after sealing, the gas was collected using a gas-tight syringe and stored in
pre-evacuated Exetainer glass bottles (Labco, High Wycombe, UK). After each sampling point, the
serum bottle was ventilated for 10 min, and then sealed with butyl rubber stoppers.

**2.5 Analytical methods**
The C content of the soil and plant residues (shoots and roots) was determined using dry combustion
with an elemental analyser (vario MAX; Elementar Analysensysteme GmbH, Germany), whereas the
$CH_4$ and $CO_2$ concentrations of the headspace samples were measured using a gas chromatographer
(Agilent 7890A, Agilent Technologies, USA) equipped with a thermal conductivity detector for
measuring $CO_2$ and a flame ionization detector for measuring $CH_4$. In addition, the stable C isotope
composition of soils and plant residues were analysed using an isotope ratio mass spectrometer coupled
with an elemental analyser (FLASH 2000; Thermo Fisher Scientific, USA), whereas the stable C
isotope composition of $CO_2$ and $CH_4$ in the headspace samples were analysed using the isotope ratio
mass spectrometer coupled with a GasBench (Thermo Fisher Scientific).

**2.6 Calculations and statistical analysis**





The $\delta^{13}C$ values of plant residues, rhizodeposits, microbe-assimilated C, soils, $CO_2$, and $CH_4$ were
calculated as follows:
$$\delta^{13}C\ (‰) = \frac{Rs - R_{PDB}}{R_{PDB}} \times 1000\ ‰, \tag{1}$$

where $R_{PDB}$ is the $^{13}C/^{12}C$ ratio of the international Pee Dee Belemnite (PDB) standard ($R_{PDB} =$
0.0112372; Lu et al., 2003) and $Rs$ is the sample $^{13}C/^{12}C$ ratio. In addition, $Atomic\ ^{13}C\ (\%)$ was
calculated as follows:
$$Atomic\ ^{13}C\ (\%) = \frac{(\delta^{13}C + 100)\ R_{PDB}}{(\delta^{13}C + 100)\ R_{PDB} + 1} \times 100\%, \tag{2}$$


and the incorporation of $^{13}C$ in plant residues, rhizodeposits, microbe-assimilated C, bulk soils, $CO_2$,
and $CH_4$ was calculated as follows:
$$^{13}C_{sample}\ (\%) = [(Atomic\ ^{13}C), L - (Atomic\ ^{13}C), UL] \times \frac{C_{sample}}{100}, \tag{3}$$

where $(Atomic\ ^{13}C),\ L$ and $(Atomic\ ^{13}C),\ UL$ are the percentages of $Atomic\ ^{13}C$ in labelled and
unlabelled samples, respectively, and $^{13}C_{sample}$ and $C_{sample}$ are the total $^{13}C$ and C content of each
sample.

199        The $^{13}CO_2$ and $^{13}CH_4$ efflux (%) were calculated as the increases in $^{13}CO_2$-C and $^{13}CH_4$-C,

respectively, as percentages of the $^{13}C$ input, within each sampling interval, whereas cumulative $^{13}CO_2$
and $^{13}CH_4$ emission (%) were calculated as the percentages of the $^{13}C$ input represented by the sum of
the $^{13}C$ in $^{13}CO_2$ and $^{13}CH_4$, respectively, at each sampling day, and the mineralization percentage of the
input $^{13}C$ was calculated as the sum of total $^{13}C$ in $CO_2$ and $CH_4$, at each sampling day, relative to the
initially added total $^{13}C$.

205        The temporal dynamics of the cumulative mineralization ratios were described by fitting a first

order single exponential decay curve:
$$y = y_0 + a\ (1 - e^{-bx}), \tag{2}$$

where $y$ is the percentage of $^{13}C$ emission from the labelled-substrate C; $y_0$ is the pool of
labelled-substrate C remaining in the soil; $a$ is the initial amount of bioavailable labelled-substrate C





pool; $b$ is the mineralization rate of substrate C; and $x$ is time (d). Mean residence time ($1/b$) and
half-life ($\ln(2)/b$) were also calculated.
The end-member mixing model was used to calculate the fractions of SOC- ($C_{SOC}$) and plant
residue-derived C ($C_{shoot}$ and $C_{root}$), as described by Phillips and Gregg (2001) and Phillips et al. (2005).
This model allows the combination of mass spectrometric and efflux measurements. The shoot-derived
$^{13}CO_2$ emission ($^{13}CO_{2shoot\text{-}derived}$) was calculated as follows:
$$^{13}CO_{2shoot-derived} = \frac{\delta^{13}CO_{2shoot} - \delta^{13}CO_{2CK}}{\delta^{13}CO_{2shoot} - \delta^{13}CO_{2soil}} \times CO_{2shoot}, \tag{5}$$

where $\delta^{13}CO_{2shoot}$ and $\delta^{13}CO_{2CK}$ are the $\delta^{13}C$ values of $CO_2$ derived from $C_{shoot}$ and $C_{CK}$, respectively;
$\delta^{13}C_{shoot}$ and $\delta^{13}C_{soil}$ are the $\delta^{13}C$ values of $C_{shoot}$ and $C_{soil}$, respectively; and $CO_{2shoot}$ is the total $CO_2$
derived from $C_{shoot}$; and the shoot-derived $^{13}CH_4$ emission ($^{13}CH_{4shoot\text{-}derived}$) and the root-derived $^{13}CO_2$
and $^{13}CH_4$ emission ($^{13}CO_{2root\text{-}derived}$ and $^{13}CH_{4root\text{-}derived}$, respectively) were calculated similarly (Phillips
and Gregg, 2001; Ye et al., 2015).
The PE of SOM on $CO_2$ and $CH_4$ emission was calculated as follows:
$$PE_t\ (\%) = \frac{Gas - Gas_{CK}}{Gas_{CK}} \times 100\%, \tag{6}$$

where $PE_t$ is the PE at time t (d); $Gas$ is the total amount of $CO_2$ and $CH_4$ derived from $C_{shoot}$ and $C_{root}$;
and $Gas_{CK}$ is the total amount of $CO_2$ and $CH_4$ derived from $C_{CK}$ (Ye et al., 2015).
Analysis of variance in conjunction with Duncan's multiple range test ($p < 0.05$) and correlation
analysis were conducted using SPSS 17 (SPSS Inc., Chicago, IL, USA), and figures were created using
Origin 8.5 (OriginLab, Northampton, MA, USA).

**3   Results**
**3.1 $CO_2$ and $CH_4$ emission of carbon substrate-treated soils**
The atomic $^{13}C$ recovered from the $CO_2$ emission of Shoot-C- and Root-C-treated soils increased
sharply at the beginning of incubation, reached a peak (4.71 and 4.39%) after 10 d, and then slowly
declined until the end of the incubation; and the atomic $^{13}C$ of $CO_2$ from Rhizo-C- and Micro-C-treated
soils exhibited a similar pattern, but with lower percentages (Fig. 1a). In addition, both Shoot-C and





Root-C had similar effects on the amount of atomic $^{13}$C recovered from $CH_4$, however, the $CH_4$ was
below the detection limit in Rhizo-C- and Micro-C-treated soils (Fig. 1b).
The $^{13}CO_2$ efflux rates also increased rapidly at the beginning of incubation, peaked after 20 d, and
then decreased gradually (Fig. 1c); however, the efflux rates from Shoot-C- (0.71%) and
Root-C-treated (0.66%) soils were higher than those of Rhizo-C- (0.11%) and Micro-C-treated (0.06%)
soils. The $^{13}CH_4$ efflux rates exhibited similar patterns (Fig. 1d). Furthermore, the cumulative $^{13}CO_2$
and $^{13}CH_4$ emissions also increased linearly during the first 60 d of incubation, after which they
increase slowly (Fig. 2). The total $^{13}$C per 100 g soil was 11.4, 5.75, 1.61, and 0.49 mg in the Shoot-C,
Root-C, Rhizo-C, and Micro-C treatments, respectively (Table 1), and the total $^{13}CO_2$ emissions
accounted for 28.6 and 43.8% of the initial $^{13}$C from Shoot-C and Root-C, respectively, and 7.9 and
7.7% of the initial $^{13}$C in Rhizo-C and Micro-C (Fig. 2a). In contrast, the cumulative $^{13}CH_4$ emissions
only accounted for 3.3 and 1.6% of the initial $^{13}$C from Shoot-C and Root-C, respectively (Fig. 2b).
The cumulative mineralization of substrate-derived $^{13}$C was more rapid at the beginning of the
incubation and followed a single exponential model (Fig. S1), and at the end of the incubation, we
found that the total mineralization percentage was highest in Root-C-treated soils (45.4%), followed by
Shoot-C- (31.9%), Rhizo-C- (7.9%), and Micro-C-treated (7.7%) soils. In addition, the bioavailable $^{13}$C
in the Shoot-C- and Root-C-treated soils was 34.2 and 46.2%, respectively, which was 4–5-fold larger
than that of the Micro-C- (9.7%) and Rhizo-C-treated (7.8%) soils, and the mean residence time (MRT)
of the Shoot-C-, Root-C-, Rhizo-C-, and Micro-C-treated soils was 33.4, 46.1, 62.9, and 192 d,
respectively (Table 2).

**3.2 Priming effect of Shoot-C and Root-C on $CO_2$ and $CH_4$ emission**
Over the entire incubation period, the cumulative emissions of $CO_2$ and $CH_4$ from the untreated soil
was 1692 mg $kg^{-1}$, and the SOC-derived C emissions from the Shoot-C- and Root-C-treated soils were
2519 mg $kg^{-1}$ and 1737 mg $kg^{-1}$, respectively, which was 1.49- and 1.03-fold that of the untreated soil
(Fig. 3). In addition, the end-member mixing model used to partition SOC-derived $CO_2$-C and $CH_4$-C
suggested that the mineralization of native SOC was promoted by the Shoot-C and Root-C treatments.
Furthermore, the PE of the Shoot-C treatment peaked at 351% after 20 d of incubation and
decreased to 46% by the end of the incubation, whereas the PE of the Root-C treatment peaked at 39%



after 50 d of incubation and then decreased to 0.8%. Thus, the positive PE of Shoot-C was clearly
stronger than that of Root-C, especially since the PE of Root-C was insignificant (i.e., $p > 0.05$; Fig. 4).

### 3.3 Mineralization of soil organic carbon in Rhizo-C- and Micro-C-treated soils

The total C emissions of Rhizo-C- and Micro-C-treated soils increased significantly from 116 mg kg$^{-1}$
and 81 mg kg$^{-1}$ after 10 d of incubation, respectively, to 1754 mg kg$^{-1}$ and 1785 mg kg$^{-1}$ by the end of
the incubation. The total C emission of Rhizo-C-treated soil was significantly higher than that of the
Micro-C- and un-treated soil, during the first 200 d of incubation; however, no significant differences
were identified at the end of the incubation (Fig. 5a). In addition, the total $^{13}$C emissions derived from
the Rhizo-C and Micro-C treated soils gradually increased over the entire incubation, and the total
mineralization of $^{13}$C in Rhizo-C-treated soils was significantly higher than that in Micro-C-treated
soils ($p < 0.05$; Fig. 5b). However, the C-mineralization of neither the Rhizo-C- nor Micro-C-treated
soils were significantly different than that of untreated soil, which suggested that the rhizodeposits and
microbe-assimilated C had effect on the mineralization of native SOC.

### 4    Discussion

### 4.1    Mineralization of carbon substrates in paddy soil

The atomic $^{13}$C and effluxes of both $CO_2$ and $CH_4$ from soils treated with $^{13}$C-labelled substrates
exhibited a rapid increase at the beginning of the incubation, followed by a slow decrease (Fig. 1),
which indicated that microbes prefer fresh C substrates over native SOC (Yuan et al., 2012c), as has
been reported by previous studies on the decomposition of fresh C substrates in both paddy and upland
soils (Lu et al., 2003; Parshotam et al., 2000). In these systems, the initial rapid decomposition is due to
the addition of easily degradable organic C, in the form of highly bioavailable compounds with low
molecular weight. Then, after the exhaustion of labile C, more recalcitrant components, such as cutin
and lignin, and mineral-stabilized SOC are utilized (Baumann et al., 2009). However, the transition
could also involve an alteration in species dominance, with rapidly proliferating bacteria using more
available compounds during the early stages of decomposition and slower-growing fungi using the
more recalcitrant components during later stages (Baumann et al., 2009; Brant et al., 2006).





Both $CO_2$ and $CH_4$ efflux are important components of the C cycle in paddy soils and represent a
major proportion of the C released by microbial decomposition (Yuan et al., 2012c), and the results of
the present study suggest that the mineralization of shoot- and root-derived $^{13}$C was ~3-4 times higher
than that of rhizodeposit- and microbe-derived $^{13}$C (i.e., Root-C > Shoot-C > Rhizo-C > Micro-C; Fig.
2). The present study also found that the percentage of Root-C-derived $^{13}$C recovered from $CO_2$ was
1.6-fold higher than Shoot-C-derived $^{13}$C, which indicated that root residue was more easily
decomposed, a conclusion that was also supported by the higher $^{13}CO_2$ efflux of Root-C-treated soils.
However, the C mineralization rates of Rhizo-C- and Micro-C-treated soils were much slower, and
the MRTs of Rhizo-C- and Micro-C-treated soils were 2–4-fold higher than those of Shoot-C- and
Root-C-treated soils, likely owing to the formation of mineral-associated organic matter during the
labelling period that was well protected from microbial degradation and had a slow turnover rate
(Basler et al., 2015; Mikutta et al., 2014; Saidy et al., 2012; Schurig et al., 2013). Furthermore, most of
the C in Rhizo-C- and Micro-C-treated soils was not mineralized to $CO_2$ but, instead, underwent
intensive internal recycling (Gunina and Kuzyakov, 2015; Knowles et al., 2010) and were stored as
living microbial biomass, and the resulting biomass was either stabilized as occluded particulate
organic matter and mineral-associated organic matter (Basler et al., 2015, Schurig et al., 2013) or was
incorporated into metabolic products, such as sugars, carboxylic acids, and amino acids, which were
incorporated into cell membranes, cell walls, or polymers (Apostel et al., 2015; Gunina et al., 2014).

## 4.2  Effect of carbon substrates on native SOC mineralization

In the present study, the emission of $CO_2$ and $CH_4$ by Shoot-C- and Root-C-treated soils during the first
50 d were mainly derived from plant residue C, after which the contribution of native SOC increased.
However, a positive PE was observed until the end of the incubation, with the exception of
Root-C-treated soils (Fig. 4). These results are supported by previous studies that have reported that the
initial phase of rapid decomposition was the result of adding easily degraded organic C and other
available nutrients that promote both microbial activity and SOC decomposition (Chen et al., 2014;).
The compounds decomposed during the slower phase were less available for microbial growth, and as a
result of C limitation, most of the available C was likely incorporated into cells and converted to
storage compounds, rather than used for growth or respiration (Lu et al., 2003; Brant et al., 2006).





However, the extracellular enzymes generated to degrade recalcitrant C substrates might be more
effective in decomposing SOC at later stages of incubation, leading to a positive PE (Chen et al., 2014).
In addition, the two phases of exogenous C decomposition and the mechanisms of PE simultaneously
influence the strength and extent of native SOC mineralization (Chen et al., 2014; Ye et al., 2015).
As sources of C that is stabilized by soil minerals, both Rhizo-C and Micro-C augmented the C
content of paddy soil (1.89 and 1.9%, respectively) over that of untreated soil (1.8%) and also reduced
native SOC decomposition, which suggests that they could be used to protect native SOC, increase the
organic carbon storage of paddy soil, and even mitigate global warming (Ge et al., 2012; Li and Yagi,
2004, Shen et al., 2014). In the present study, we found that the C emissions of Rhizo-C- and
Micro-C-treated soils were similar to those of untreated soil, which suggested that rhizodeposits and
microbe-assimilated C input have no effect on native SOC mineralization at any time during the
rice-growing season. However, the total Rhizo-C- and Micro-C-derived $^{13}$C increased gradually over
the incubation period, despite the small initial input during the 18-d labelling period, which implies that
the soils' native SOC-derived C were smaller than those of untreated soil and that both Rhizo-C and
Micro-C suppressed the mineralization of native SOC. This suppression probably occurred because
rhizodeposits and microbe-assimilated C are predominantly composed of highly bioavailable
compounds with low molecular weight (Lu et al., 2002) and, in the present study, likely underwent
internal recycling *via* microbial metabolism during the incubation, as indicated by their relatively
longer MRTs (Table 2; Gunina et al., 2014; Mikutta et al., 2014; Schurig et al., 2013).

## 5    Conclusions

In the present study, Root-C-treated soils exhibited the highest rate of C mineralization, followed by
Shoot-C-, Rhizo-C-, and Micro-C-treated soils, whereas the opposite trend was observed for MRT, and
by the end of 300-d incubation, both Shoot-C- and Root-C-treated soils exhibited higher total
mineralization and positive PEs. Although plant residues are widely used for improving soil fertility,
their contribution to SOC assimilation is inefficient, and their use also contributes to the emission of
greenhouse gasses. However, the present study demonstrates that both rhizodeposits and
microbe-assimilated C can reduce native SOC decomposition and may more effectively contribute to
the stability and sequestration of soil C.






*Acknowledgments*. The present study was supported by the National Natural Science Foundation of
China (41430860, 41371304), the Strategic Priority Research Program of the Chinese Academy of
Sciences (XDB15020401), the Open Foundation of Key Laboratory of Agro-ecological Processes in
Subtropical Region, the Chinese Academy of Sciences Institute of Subtropical Agriculture
(ISA2015101), and the Recruitment Program of High-End Foreign Experts of the State Administration
of Foreign Experts Affairs, awarded to Prof. Georg Guggenberger (GDT20154300073).



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





**Tables**

**Table 1.** The carbon (C) content, atomic $^{13}$C, and total $^{13}$C in the soil and photosynthesized C substrates after 18 d of $^{13}$C-labelling.

|  | Bulk soil | Shoot-C | Root-C | Rhizo-C | Micro-C |
|---|---|---|---|---|---|
| C content (%) | 1.80 ±0.12 | 40.50 ±2.13 | 28.60 ±1.15 | 1.89 ±0.12 | 1.90 ±0.11 |
| Atomic $^{13}$C (%) | 1.08 ±0.02 | 5.78 ±0.09 | 4.43 ±0.07 | 1.16 ±0.03 | 1.11 ±0.02 |
| Total $^{13}$C (mg) | 0 | 11.43±0.52 | 5.75 ±0.41 | 1.61 ±0.06 | 0.49 ±0.05 |

Bulk soil, unplanted and unlabelled soil; Shoot-C, paddy soil supplemented with $^{13}$C-labelled shoot residue; Root-C, paddy soil supplemented with $^{13}$C-labelled root residue; Rhizo-C, paddy soil supplemented with $^{13}$C-labelled rhizodeposits; Micro-C, paddy soil supplemented with $^{13}$C-labelled microbe-accumulated C.





**Table 2.** The size of bioavailable labelled-substrate C pool, mean residence time (MRT), and half-life
of cumulative $^{13}C$ recovery in $CO_2$ and $CH_4$ in four different incubation treatments.

| Treatment | Size (%) | MRT (d) | Half-life (d) | $R^2$ |
|-----------|----------|---------|---------------|-------|
| Shoot-C   | 34.2     | 33.4    | 23.2          | 0.99  |
| Root-C    | 46.2     | 46.1    | 31.9          | 0.99  |
| Rhizo-C   | 7.8      | 62.9    | 43.6          | 0.98  |
| Micro-C   | 9.7      | 192     | 133           | 0.98  |

The size of bioavailable labelled-substrate C pool (% initial $^{13}C$), MRT, and $R^2$ were calculated based
on Fig. 1S. Shoot-C, paddy soil supplemented with $^{13}C$-labelled shoot residue; Root-C, paddy soil
supplemented with $^{13}C$-labelled root residue; Rhizo-C, paddy soil supplemented with $^{13}C$-labelled
rhizodeposits; Micro-C, paddy soil supplemented with $^{13}C$-labelled microbe-accumulated C.





**Figures captions**
**Figure 1.** Atomic $^{13}C$ (%) recovered from $CO_2$ (**a**) and $CH_4$ (**b**) emissions and the $^{13}CO_2$ (**c**) and $^{13}CH_4$
(**d**) efflux (% initial $^{13}C$) over the 300-d incubation period. The inset in Fig. 1**c** shows the $^{13}CO_2$ efflux
(% initial $^{13}C$) of Rhizo-C and Micro-C over the 300-d incubation period. Values and error bars
represent means $\pm$ SE (n = 4). Shoot-C, unlabelled paddy soil supplemented with $^{13}C$-labelled shoot
residue; Root-C, unlabelled paddy soil supplemented with $^{13}C$-labelled root residue; Rhizo-C, paddy
soil containing $^{13}C$-labelled rhizodeposits; Micro-C, paddy soil containing $^{13}C$-labelled
microbe-accumulated C.

**Figure 2.** Cumulative $^{13}CO_2$ (**a**) and $^{13}CH_4$ (**b**) emissions (% of initial $^{13}C$) over the 300-d incubation
period. Values and error bars represent means $\pm$ SE (n = 4). Shoot-C, unlabelled paddy soil
supplemented with $^{13}C$-labelled shoot residue; Root-C, unlabelled paddy soil supplemented with
$^{13}C$-labelled root residue; Rhizo-C, paddy soil containing $^{13}C$-labelled rhizodeposits; Micro-C, paddy
soil containing $^{13}C$-labelled microbe-accumulated C.

**Figure 3.** Cumulative $CO_2$ and $CH_4$ emissions by Shoot-C- (**a**) and Root-C-treated (**b**) soils over the
300-d incubation period. The inset in Fig. 3**b** shows total C emission derived from SOC of
Root-C-treated soils and CK. Values and error bars represent means $\pm$ SE (n = 4). Shoot-C, $^{13}C$-labelled
shoot residue; Root-C, $^{13}C$-labelled root residue; SOC, soil organic carbon; CK, unlabelled and
unplanted soil without supplementation.

**Figure 4.** Priming effect (%) of $^{13}C$-labelled plant residues over the 300-d incubation period. Values
and error bars represent means $\pm$ SE (n = 4). Shoot-C, unlabelled paddy soil supplemented with
$^{13}C$-labelled shoot residue; Root-C, unlabelled paddy soil supplemented with $^{13}C$-labelled root residue.

**Figure 5.** Total C (**a**) and $^{13}C$ (**b**) emission by $^{13}C$-labelled rhizodeposit- and microbe-accumulated
C-treated soils over the 300-d incubation period. Values and error bars represent means $\pm$SE (n = 4).
Different letters indicate significant differences at $p < 0.05$ (Duncan multiple range test). Rhizo-C,



paddy soil containing $^{13}$C-labelled rhizodeposits; Micro-C, paddy soil containing $^{13}$C-labelled
microbe-accumulated C; CK, unlabelled and unplanted soil without supplementation.





**Figure 1**

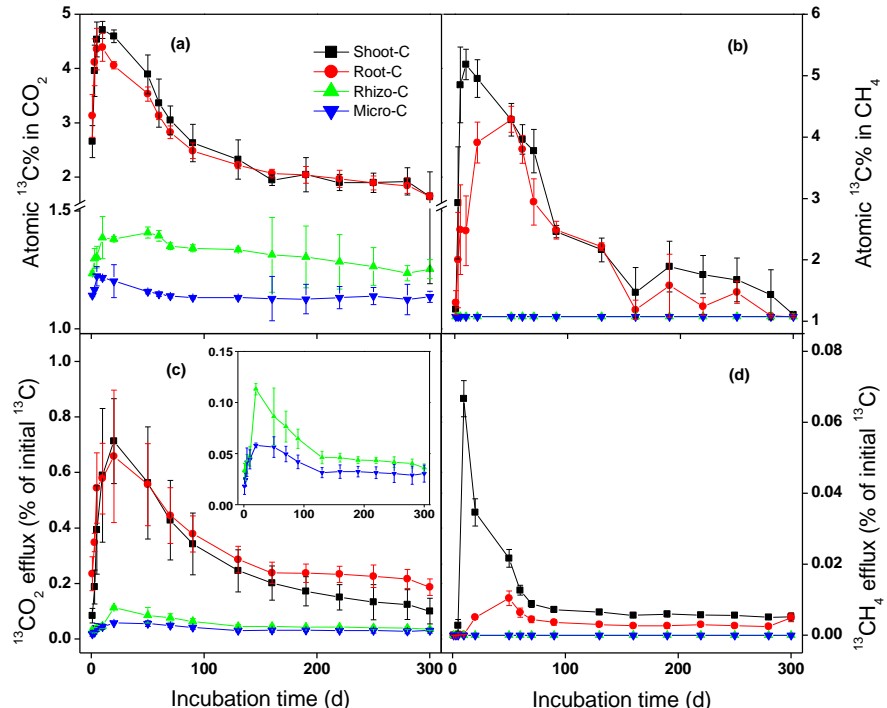







**Figure 2**

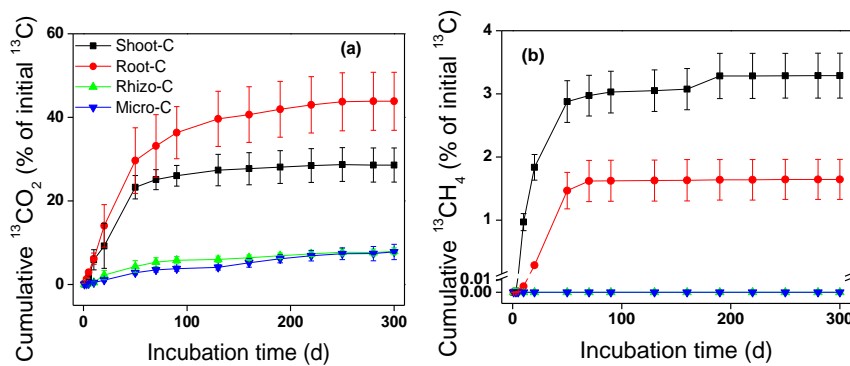







**Figure 3**

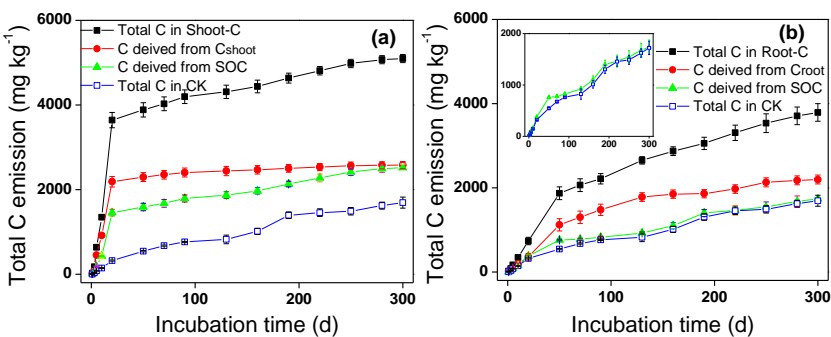





**Figure 4**

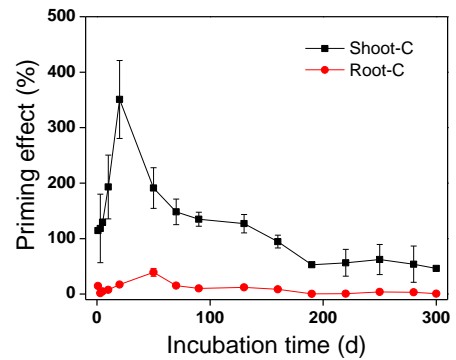







**Figure 5**

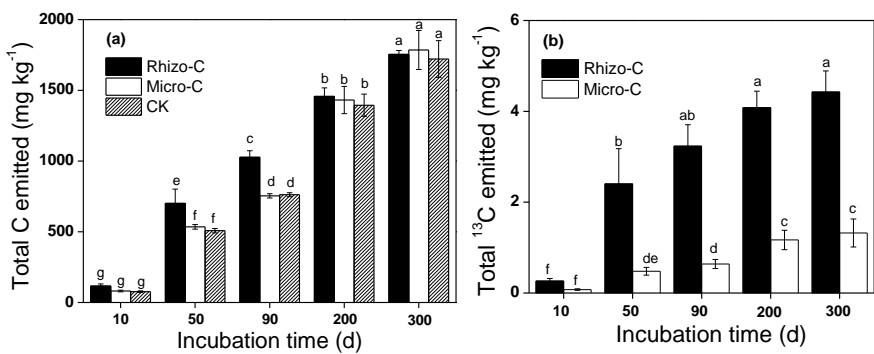
