# Peer review of "Fate of rice shoot and root residues, rhizodeposits, and microbe-assimilated"

_Biogeosciences, 2016_

## Referee Comment (RC1) · Anonymous Referee #1 · 17 May 2016

General comments The manuscript submitted by Zhu et al. investigated the fate and priming effect of organic C incorporated into paddy soils by plants and microorganisms through a long incubation study. The four different 13C-labelled substrates, i.e., rice roots, shoots, rhizodeposits and microbial assimilated carbon were used to analysis the mineralization processes. The topic is within the scope of this journal, and benefit for understanding the dynamics of C cycling and environmental protect in paddy soil ecosystem. They found that the mineralization of native soil organic matter was 2.6 and 2.0 fold higher in shoot-C and root-C, and positive priming effects were found in the two C substrates. The rhizodeposits and microbial assimilated C showed no

significant differences in the total amount of emitted carbon compared with control and the negative priming effects. The above results can give new insights for soil C turnover and biogeochemical cycling mechanisms. In summary, this is a strong and well-done manuscript that needs only minor revisions.

Specific comments

1. The incubation study used 13C labeled microbial assimilated C and rhizodeposited C. How to label them? Please give more details about this.

2. The study is a long incubation experiment. How to reduce the cross-feeding effect? Especially, all the rice shoots, roots, rhizodeposits can be assimilated into microbial biomass C. Did the formulas already take into account the cross-feeding effects between different C substrates?

---

## Referee Comment (RC2) · Anonymous Referee #2 · 22 May 2016

Zhu et al. present an interesting and cleverly designed experiment examining the fate of different 13C-labeled tissues in rice paddy soil. However, in my opinion there are several important deficits in the presentation of the manuscript, one potentially significant error in analysis, and critical caveats of interpretation that should be considered.

First, the authors use an isotope mixing model based on delta notation (line 216) to partition $CO_2$ from native SOM vs. 13C-labeled tissues. This is likely to yield biased results, because delta notation becomes highly nonlinear with respect to 13C atom percent away from zero per mil, and thus delta notation should not be used in the

context of isotope labeling experiments. This flaw can be readily fixed by using atom percent 13C values, as opposed to delta values, in the mixing model. This should affect the magnitude of calculated C fluxes, but not the direction of the results.

Second, there appears to be confusion and a misstatement with respect to the total amount of C added in each treatment. If I understand correctly based on the methods, the total C in each treatment is as follows: Shoot C and Root C treatments should have 100 g of bulk soil (1.8 % C) plus 0.6 g tissue with C content of 41 and 29 %, respectively. This yields total mass-weighted C content of 2.04 % and 1.97 % based on the above data, which is in fact substantially greater than the other treatments (1.9%), in contrast to what is claimed without support in the abstract (where it is claimed that the Rhizo C and micro C had greater C).

This difference in C inputs among treatments is important to consider in the context of priming, on of the main foci of the study. If one assumes that there is a limited and finite capacity for stabilization of fresh C inputs to soil, regardless of source, one might postulate that the priming response to addition of C varies with the amount of C added. Thus, one could potentially observe differences in priming among treatments simply due to C quantity, in addition to the likely impact of biochemical differences among C substrates. This is especially important given that the treatment which exhibited the greatest priming also had the greatest C addition (2.04% for shoot C). I don't think this is necessarily a fatal flaw, but rather an important limitation of interpretation that needs to at least be acknowledged and discussed. It seems odd to me that the experiment was not designed to add a uniform amount of organic matter among treatments.

Third, estimates of variability around means are typically not presented. These are critically needed to interpret differences (or lack thereof) among treatments. There is also confusion and contradiction in the manuscript about which differences are significant or not, particularly with respect to priming in the root addition treatment (once it is stated that there was a positive priming effect, elsewhere it is stated that this was not significant). These will likely need to be re-evaluated with the new mixing model results

from atom percent data, as discussed above.

Fourth, the hypothesis that was posed at the end of the introduction was ambiguous and was not further addressed in the discussion. The hypothesis needs to be justified in the Introduction, and evaluated in the context of the data in the Discussion.

Fifth, because there was a different mass of 13C added to each treatment, I cannot see how Figure 1a,b are useful, and these should likely be removed—Figures 1c,d show the normalized data and are much more useful.

Sixth, Tables 1 and 2 are confusing and possibly contain errors, as discussed below. Finally, although I agree with the authors' overall interpretation of the data, there are several sentences that are logically inconsistent throughout the manuscripts, where the statement at the beginning of the sentences does not support what follows. There is also substantial speculation and extraneous text that should be revised or removed. These are detailed below.

Detailed comments: 32-36: This statement is not logically consistent. An increase in 13C emissions does not imply lower soil organic C decomposition, nor that the rhizo-C and micro-C soils decrease mineralization of native soil C. 52: Heterotrophic microbes are typically much more abundant in terms of biomass than autorophs, and would be expected to be a more important C input to SOM. This distinction is not important here. 58-59: But you just mentioned the importance of microbes. . . green manure and manure are also often used in paddy systems. 75-77: This may be statistically significant but autotrophic microbial C fixation is equivalent to a rounding error in the total C budget of these systems. . . 81-83: This is a false dichotomy, as plant residues decompose to yield low molecular weight substances. 88: Contradicts the above statement, where you asserted that straw leads to priming. 97: How do you define complexity here? It is unclear whether fresh plant tissue or microbial biomass would be more complex than the other in terms of biochemical composition. This hypothesis needs to be introduced and justified in the context of the literature. 127: I disagree with this statement—microbes were definitely exposed to the 13CO2 label given that root respiration would have been enriched in 13C. Even heterotrophic microbes assimilate CO2 via anapleurotic fixation. This does not matter in the context of your treatments, and this text could be removed. 215: Delta notation should not be used for 13C-enriched samples because it is highly nonlinear away from 0 permil. The mixing analysis should be repeated using the atom percent data. 235: This contradicts what was stated in the abstract with respect to trends in 13C in the rhizo C and Micro C treatments. 232-237: Because there was a different mass of 13C label added to each treatment I think that Figure 1 a,b is misleading. Figure 1 c,d normalize the 13CO2 fluxes to the amount of label added, thus the treatments can be readily compared. I recommend removing Fig. 1a,b and the associated text in the Results. 250-255: Standard errors associated with these percentages are needed. 259-260: Standard errors needed 273-274: Isn't it trivial that the cumulative 13CO2 respired increased over the experiment? Discussing rates of change would be more informative. 278: Do you mean "no" effect? 304-310: This claim cannot be supported by the present data, and should be couched as speculation or removed. 319-321: Unsupported speculation 322-323: But you saw PE decrease over time, right? 328: But in natural systems, Rhizo C and Micro C typically accompany root and shoot C—they are not present on their own, unless roots and shoots are manually removed. One implication of your results might be that soil C would disproportionately benefit from shoot removal by farmers—is this correct? 330-332: Better support for this claim would come from the isotope mixing model. 333: I assume you mean 13C of CO2? Need to specify here and elsewhere. 333: Unnecessary to include "rice-growing season" given that this is not a field study. 333-336: This conclusion does not follow from the premise. This sentence is confusing and not logically consistent. 336-340: That is one hypothesis; another would be that these tissues are selectively stabilized due to interactions with minerals or aggregate formation. This uncertainty should be acknowledged. 346: You stated before that the PE for root treated soils was insignificant. Need to be consistent in the text—is it significant or not? If not, PE is not positive. 350: Should mention as a caveat that different amounts

[Figure]

of C were added in each treatment, and it is uncertain whether this contributed to differences in the results. Table 1: The third row is unclear—why does bulk soil have 0 mg total 13C, when it is 1.08 atom percent 13C? You need to clarify or account for natural abundance 13C. Table 2: "Size" of the pools is unclear here—is this the proportion of 13C that was respired over the experiment?
* * *

---

## Author Comment (AC1) · 23 May 2016

Dear Anonymous Referee #1, Thank you very much for your valuable comments. We would like to answer your concerned points one by one (Q, plain, and A, blue font).

Q1. General comments. The manuscript submitted by Zhu et al. investigated the fate and priming effect of organic C incorporated into paddy soils by plants and microorganisms through a long incubation study. The four different 13C-labelled substrates, i.e., rice roots, shoots, rhizodeposits and microbial assimilated carbon were used to analysis the mineralization processes. The topic is within the scope of this journal,

and benefit for understanding the dynamics of C cycling and environmental protect in paddy soil ecosystem. They found that the mineralization of native soil organic matter was 2.6 and 2.0 fold higher in shoot-C and root-C, and positive priming effects were found in the two C substrates. The rhizodeposits and microbial assimilated C showed no significant differences in the total amount of emitted carbon compared with control and the negative priming effects. The above results can give new insights for soil C turnover and biogeochemical cycling mechanisms. In summary, this is a strong and well-done manuscript that needs only minor revisions.

A: Thank you for the positive comment.

Q2. The incubation study used 13C labeled microbial assimilated C and rhizodeposited C. How to label them? Please give more details about this.

A: To obtain the 13C labeled microbial assimilated C and rhizodeposited C we firstly labelled the rice and soil, then collected the soils containing microbial assimilated C and rice rhizodeposits. Namely: "Rice cultivation and 13CO2 labelling were performed as described by Ge et al. (2012; 2013), with some modifications. For 13C labelling, 30 pots (20 planted, 10 unplanted) were transferred to an automatically controlled gas-tight growth chamber (110 cm length, 250 cm width, 180 cm height) and exposed to 13CO2-fumigation for 18 d (May 14–31, 2013), during the vegetative growth period (including the entire tillering stage). The growth chambers were placed in a rice field to ensure that the environmental conditions of the labelled. The surface of each planted pot was covered with black plastic sheeting, to prevent algal photosynthesis in the floodwater and to ensure that only the rice shoots were exposed to 13CO2 (i.e., not phototrophic microbes in the soil or water), whereas the unplanted pots were left uncovered, so that the soils were directly exposed to 13CO2 and so phototrophic soil microbes could assimilate atmospheric 13CO2. All the pots were watered every few days, in order to maintain a water depth of 2–3 cm above the soil surface, until harvest. The CO2 concentrations of the growth chambers were measured using an infrared analyser (Shsen-QZD, Qingdao, China) and maintained at 360–380 $\mu$l L-1. The 13CO2

was generated by acidifying Na2 13CO3 (1.0 M, 99 atom % 13C; Cambridge Isotope Laboratories, Tewksbury, MA, USA) with H2SO4 (0.5 M) in beakers that were placed inside the growth chambers. During the labelling period, 13CO2 was only released when CO2 concentrations fell below 360 $\mu$l L-1, and at CO2 concentrations >380 $\mu$l L-1, the gas flow was diverted and passed through CO2 traps (NaOH solution). An air-conditioning system was used to control the temperature inside the chamber within 1 °C of the ambient temperature in the rice field. Two fans continuously circulated the air in the growth chamber. The soils were sampled destructively after 18 d of 13CO2 labelling. 13C-labelled rhizodeposits were obtained by gently shaking moist soil from the roots of rice plants and were prepared for incubation by removing root debris and mixing thoroughly. Meanwhile, to obtain microbe-assimilated 13C, we collected soil from 13C-treated, unplanted pots and mixed it thoroughly."

Q3. The study is a long incubation experiment. How to reduce the cross-feeding effect? Especially, all the rice shoots, roots, rhizodeposits can be assimilated into microbial biomass C. Did the formulas already take into account the cross-feeding effects between different C substrates?

A: Many thanks for the kind comments. We acknowledged that the cross-feeding effects occurred in our experiment, but we were focusing on the net effect of the different substrates (primers) on the decomposition of native soil organic carbon in our present study. We can further work on the cross-feeding effect as your argument.

---

## Referee Comment (RC3) · Anonymous Referee #3 · 3 Jun 2016

This paper presents an incubation experiment using rice tissues and soils labeled with 13C. Labeled shoots and roots were directly added to soil. Rhizodeposits were added by shaking soil from roots, and microbe-fixed C was added by using soils that were sunlit and treated with 13C but had no plants.

This study addressed important issues related to priming effects in rice agriculture, and is appropriate for this journal. The results have important implications for C emissions, soil carbon storage, and potential strategies for mitigating greenhouse gas emissions from agriculture. The isotope labeling procedure and the incubation were sound.

However, there were some issues with the analysis and interpretation that should be addressed.

1. In general, the root and shoot amendments seem sound. However, the amount of labelled carbon added in the Rhizo-C and Micro-C treatments was much smaller than in the root and shoot treatments, and it's not clear whether the different treatments can be directly compared with each other. Judging from Table 2, the amount of 13C added to the soil in the Rhizo-C and Micro-C treatments was very small relative to the ambient 13C content of the soil, and I am not convinced that the emissions from these small additions were enough to be detectable in this experiment. It's difficult to tell how much labeled substrate was actually added in those two treatments, and in my opinion it calls the interpretation of results related to those additions into question. I think there should be more discussion of why the 13C emissions from these treatments can be interpreted as resulting from the amendments rather than just mineralization of ambient 13C that was already present in the soils.

2. There are issues with the equations. Most of them have typographical errors or confusing notation.

3. The calculation of priming effects is problematic. They are defined using cumulative emissions. However, they are then interpreted as changes over time with statements like "a positive priming effect was observed until the end of the incubation." If calculated using cumulative emissions, any short-term priming effect would appear to last for the entire experiment, because the additional emissions at the beginning would not be cancelled out by any negative emissions later in the experiment (unless there are negative priming effects later on). Cumulative emissions could be used to calculate a total priming effect over the entire experiment in terms of extra carbon lost from SOC, but a time series of fractional priming effects like the results presented here would make much more sense if it were calculated using emission rates rather than cumulative emissions.

Specific comments follow below:

Line 32-36: I don't follow the logic of this statement. According to Fig. 1, emission rates from Rhizo-C and Micro-C were decreasing over most of the incubation. Cumulative 13C emission increased over the experiment, of course, but this only means that emission rates were greater than zero.

Line 75-77: These are tiny fractions. Are they really detectable in this kind of experiment? It's a factor of 100 less than rhizo-deposits.

Line 79-81: There is a balance between microbial decomposition and mineral sorption of these substrates, and there's a lot of uncertainty about how much is respired vs sorbed over various time scales. This balance probably depends on soil physical and chemical factors, and might be different in frequently flooded soils.

Line 97-98: There aren't really any measurements of the "complexity of substrate composition" (which isn't clearly defined either)

Line 98-99: This sentence isn't very clear. Is "their relatively higher quantity and stability in soil" referring to plant residues or rhizodeposits and microbe-assimilated C? Shouldn't substrates with higher stability in soil cause weaker priming effects, because they are more resistant to decomposition?

Line 144-145: The procedures for collecting rhizodeposits and microbe-assimilated 13C sound like they include a little bit of labeled material mixed with a lot of soil, which means these additions were quite different from the plant tissue amendments, which were pure labeled tissue. This raises questions about whether the rhizodeposit and microb-assimilated C additions are directly comparable to the root and biomass additions.

Line 164-167: The amount of carbon in these two treatments is not well known, and likely very different from the other two treatments, making direct comparison tricky.

Equation 2: I'm not an expert on isotope labeling math, but this equation looks a little

strange. What is ($\delta$13C+100) doing? I think it should be ($\delta 13C + 1000$), which equals $R_S/R_{PDE}$. Either way, it seems needlessly confusing to convert $R_S$ into per-mil units and then convert that into atomic percent, instead of just using $R_S/(R_S+1)$, which as far as I can tell is mathematically equivalent.

Equation 3: The notation of this equation (with all the brackets and commas) is confusing. It would be easier to read with some different notation (subscripts or something).

Line 207: This is labeled equation 2 but should be equation 4. It also doesn't make sense relative to the description on lines 208-210. If y is a percentage of 13C emission, then all of the terms in the equation should be percentages, while in fact they are pools. If $y_0$ is the pool of labeled C remaining in the soil, then it should be decreasing with time. The description of $a$ is basically the same as $y_0$. This equation would make more sense (relative to the description of the terms) if it were $y_0 = a(1 - e^{-bx})$.

Equation 5: Should the denominator have $\delta 13C_{shoot}$ and $\delta 13C_{soil}$ rather than $\delta 13CO2_{shoot}$ and $\delta 13CO2_{soil}$? $\delta 13CO2_{soil}$ in the equation doesn't seem to be a thing that was actually measured.

Equation 6: Priming effects are defined here as the difference in total C emissions between the amended experiments (Cshoot or Croot) and the control experiment (CK). This includes the C emissions from the decomposition of the added material as well as extra decomposition of native SOC. This is not how priming effects are usually defined, or discussed in the introduction. Usually, priming effects are defined as extra decomposition of just the native SOC, excluding emissions from the added material. If that's the case, then this equation should be isolating emissions derived from native SOC rather than using total emissions. Also, I think it would make more sense to compare emission rates rather than cumulative emissions in this ratio. If the priming effect occurs as a short pulse effect, then using this equation will exaggerate how long the priming effects last, because the increase in cumulative emissions will slowly decline as it's divided by increasing total emissions, even after increases in emissions

due to priming have ceased.

Lines 239-241: These units don't make sense for emission rates, unless they are percent of initial 13C lost over a specified time period (% per day or something). It's hard to interpret this without knowing what the initial 13C was for each treatment. Those values are in Table 1, and it would help to discuss those before going into percentage losses.

Lines 245-246: Based on Table 1, the initial 13C in rhizo-C and Micro-C treatments is nearly indistinguishable from the unlabeled bulk soil value. Are these measurements sensitive enough to determine how much of these 13C emissions was from the labeled amendments of those treatments and how much was from the ambient 13C content of the soil?

Line 248-249: Why is this figure in supplemental material instead of main text?

Line 259: The methods don't describe exactly how SOC-derived C emissions were calculated. Is this the root or shoot-derived CO2 emission from equation 5 subtracted from total emission?

Line 269: Of course the total C emissions increased. It would be impossible for them to decrease unless C emissions were negative.

Line 271-273: The Rhizo-C treatment included an addition of soil that was shaken off roots, so there was extra carbon. This might explain the greater total C emission.

Line 276-278: This could just be because the amount added was very small compared to the amount of root and shoot material added.

Line 278 should say "had no effect on the mineralization. . .".

Also, why do section 3.3 and the associated figure only address two of the four treatments?

Line 295-297: It's hard to compare these directly since different amounts of labeled

substrate were added. Based on Table 1, most of the 13C in the Rhizo and Micro treated soil could have been native SOC, and so it's not surprising that the total emissions would be small relative to the initial total.

Line 302-303: There is no evidence in this study to support this statement about mineral-associated organic matter. It's a possible explanation, but shouldn't be presented as a finding supported by this experiment.

Line 305-310: There also doesn't seem to be any evidence to support any of these statements. There weren't any measurements of 13C incorporated into microbial biomass, or any of the compounds listed in line 309, so it shouldn't be stated as something found in this experiment.

Line 315: It's misleading to say that a positive PE was observed until the end of the incubation, because the PE was calculated using cumulative values. The only way the PE could stop being positive would be if there were a negative effect on total emissions later in the incubation that reduced the cumulative emissions of the amended soils.

Also, it's misleading to say "with the exception of the Root-C-treated soils", since only two things are being compared. "Exception" implies that only one thing was different out of a larger group.

Line 322-323: Again, this statement isn't supported by PE calculated using cumulative emissions. If there is a PE observed using instantaneous emissions, it might be a more reasonable explanation.

Line 327: Those differences were not statistically significant and very small, so I don't think this statement is really supported by the evidence. Certainly not enough to make such a strong statement about using them to increase SOC and mitigate global warming without stronger evidence.

Line 331-333: This was true for this study, but the amendments were very small. Maybe larger amendments would cause stronger effects?

Line 333-334: Since cumulative emissions are being shown, an increase in 13C is guaranteed as long as emissions are greater than zero, so this doesn't prove much. I don't follow the connection with suppression of native SOC mineralization. Given that SOC contained some amount of ambient 13C, I'm not sure this result says anything about the treatment effect on SOC.

Line 336-340: These statements are not really supported by any evidence from the experiment.

Table 1: Total 13C should be in mg per some mass of soil, not just mg. Bulk soil total 13C shouldn't be zero – those soils have 1% atomic 13C and nonzero C content, so they must contain some 13C as well. In fact, based on the numbers the amount of 13C in the Micro-C and Rhizo-C should be very difficult to distinguish from the amount of 13C in unlabeled soil.

Figure 1: Panels (c) and (d) show emissions in units of % of initial 13C. These units don't make sense for emission rates, unless they are percent of initial 13C lost over a specified time period (% per day or something).

Figure 3: The legend is confusing because it uses "Total C in . . ." and "C derived from . . ." to refer to the same thing (i.e. cumulative C emissions). Also, "derived" is misspelled.

---

## Short Comment (SC1) · 6 Jun 2016

This manuscript aimed to show how different C supplied from rice plant and soil microbes promoted C mineralization and interaction with native C in incubated experiment, which would be useful information to assess soil C sequestration and greenhouse gas mitigation in paddy soil, because the knowledge of C dynamics in waterlogged paddy soil is still limited to compare with aerobic upland soil (unlike L93). However one critical point of methodology is the way of preparing microbe-assimilated $^{13}C$ in soil without rice plant (L146); it is well-known that microbial activity is higher with rice

plant than without plant, in general, so that the results of microbe-assimilated 13C in this investigation might be underestimated. Also rhizodeposits (L144) were obtained by gentle sharking moist soil, which is common method for aerobic soil, but doubt for wet paddy soil, resulting again underestimate the contribution in 13C dymanics.

Other minor points are as below to improve the manuscript.

L31; Effective digits are not uniformed as 1.89 vs 1.9 and 1.8. Also L327.

L83; Yuan et al. 2014c, but only one Yuan et al. for 2014 in the list. Also L294.

L112; pH of 5.6 and a soil: water .. should be pH of 5.6 at a soil: water . . . . . .

L137; how about humidity? It is important to regulate photosynthesis/respiration.

L163; if the same bottle for the almost the same soil/water contents, water level should be not as in such wider range < 1-2 cm.

L167; how to adjust gas pressure during incubation especially with plant added?

L195 C sample; unit is not clear.

L242; linearly should be exponentially?

L249; Fig S1, missing or not available.

L277; different that .. should be different from ..

L291 slower-growing fungi; Is this applicable/useful reference to paddy soil?

L318; remove ; after Cheng et al, 2014

Table 1; Why zero for Total 13C of Bulk soil? Negligible 13C natural abundance?

---

## Author Comment (AC2) · 12 Jul 2016

Dear Anonymous Referee #2,

Thank you very much for your valuable comments. We would like to answer your concerned points one by one (Q, plain, and A, blue font).

Q1. Zhu et al. present an interesting and cleverly designed experiment examining the fate of different $^{13}$C-labeled tissues in rice paddy soil. However, in my opinion there are several important deficits in the presentation of the manuscript, one potentially significant error in analysis, and critical caveats of interpretation that should be considered.

A: Thanks for your work on our paper and raising important caveats. We have revised the manuscript accordingly.

Q2. First, the authors use an isotope mixing model based on delta notation (line 216) to partition $CO_2$ from native SOM vs. $^{13}$C-labeled tissues. This is likely to yield biased results, because delta notation becomes highly nonlinear with respect to $^{13}$C atom percent away from zero per mil, and thus delta notation should not be used in the context of isotope labeling experiments. This flaw can be readily fixed by using atom percent $^{13}$C values, as opposed to delta values, in the mixing model. This should affect the magnitude of calculated C fluxes, but not the direction of the results.

A: We have revised the mixing model equation and recalculated it using atom percent $^{13}$C values instead of delta values in the mixing model. See details in P 9-10, L 213-223.

Q3. Second, there appears to be confusion and a misstatement with respect to the total amount of C added in each treatment. If I understand correctly based on the methods, the total C in each treatment is as follows: Shoot C and Root C treatments should have 100 g of bulk soil (1.8 % C) plus 0.6 g tissue with C content of 41 and 29 %,

respectively. This yields total mass-weighted C content of 2.04 % and 1.97 % based on the above data, which is in fact substantially greater than the other treatments (1.9%), in contrast to what is claimed without support in the abstract (where it is claimed that the Rhizo C and micro C had greater C).

A: Yes, the total amount of C in the treatment of Shoot C and Root C was 2.04 % and 1.97 %, respectively. However, in the abstract we compared the total C amount of Rhizo-C- and Micro-C-treated soils with the untreated soil, but we did not compare them with Shoot C and Root C treated soil. It reads "the total C contents of Rhizo-C (1.89%) and Micro-C treated soils (1.90%) were higher than those of untreated soil (1.81%)".

Q4. This difference in C inputs among treatments is important to consider in the context of priming, one of the main foci of the study. If one assumes that there is a limited and finite capacity for stabilization of fresh C inputs to soil, regardless of source, one might postulate that the priming response to addition of C varies with the amount of C added. Thus, one could potentially observe differences in priming among treatments simply due to C quantity, in addition to the likely impact of biochemical differences among C substrates. This is especially important given that the treatment which exhibited the greatest priming also had the greatest C addition (2.04% for shoot C). I don't think this is necessarily a fatal flaw, but rather an important limitation of interpretation that needs to at least be acknowledged and discussed. It seems odd to me that the experiment was not designed to add a uniform amount of organic matter among treatments.

A: Yes, we acknowledge that the amount of C added to soil is an important factor affect priming effect. In our study, the amount of plant residues C added to soil was based on the fertilizing amount in field, the straw and root were added at a rate of 6 g/kg. As the amount of rhizodeposited C and microbial assimilated C are significantly lower than plant residues C in natural environment, we have also chosen for a smaller rate in our

experiment. In order to evaluate the effect of different photosynthetic C sources during the simulated fallow period on native SOC mineralization under realistic conditions, we didn't add a uniform amount of organic matter among treatments. But we got your point that this complicates direct comparison of the carbon sources. In the discussion we addressed this complication at P.14, L.340-342, where we stated the following: "Besides the lower contents of Rhizo-C and Micro-C as compared to Shoot-C and Root-C, this observation is possibly due to the different behaviour of primers in soil ".

Q5. Third, estimates of variability around means are typically not presented. These are critically needed to interpret differences (or lack thereof) among treatments. There is also confusion and contradiction in the manuscript about which differences are significant or not, particularly with respect to priming in the root addition treatment (once it is stated that there was a positive priming effect, elsewhere it is stated that this was not significant). These will likely need to be re-evaluated with the new mixing model results from atom percent data, as discussed above.

A: The variability of replicates was considered in the revised version, and we have added the standard errors. We have also revised the mixing model equation and recalculated PE. We obtained the results that shoot and root addition increased C emission up to 11.4 and 2.3 times higher than that of the control soil by day 20, respectively, the stimulatory effect persisted to the end of incubation period in case of Shoot-C. Over the entire incubation, the priming effect of Shoot-C on $CO_2$ and $CH_4$ emission was strongly positive over the entire incubation, however, Root-C failed to exhibit a significant positive priming effect.

Q6. Fourth, the hypothesis that was posed at the end of the introduction was ambiguous and was not further addressed in the discussion. The hypothesis needs to be justified in the Introduction, and evaluated in the context of the data in the Discussion.

A: Thank you very much for this important comment. You are right, the hypothesis is ambiguous and not well prepared for in the Introduction. In the revised version we hope that this has been improved. The hypothesis chapter is rewritten at P.5-6, L.100-105 as follows: "There were only limited studies of estimating the fate of plant residues and rhizodeposits in paddy soils and, to our knowledge, there is no comparative information on (1) the decomposition of different organic C sources, such as rice shoots and roots, rhizodeposits, and microbe-assimilated C; or (2) the effects different organic C sources on the mineralization of native SOC. We hypothesized that depending of the type of the primer both, the decomposition of the primer itself and with that the PEs on native soil organic matter vary. We assume that shoots and roots, entering the soil as unprotected particulate organic residues, are well available for microorganisms and thus also stimulate native organic matter decomposition. In contrast, rhizodeposits and microbial carbon reflect a carbon sources that are rather stabilized and contribute less to priming. We investigated these hypotheses by quantifying the contribution of different organic C sources to $CO_2$ and $CH_4$ emission and by analysing their PE, in a 300-d incubation study using $^{13}C$-labelled rice plant residues, rhizodeposits, and microbe-assimilated C in paddy soils."

Q7. Fifth, because there was a different mass of $^{13}C$ added to each treatment, I cannot see how Figure 1a,b are useful, and these should likely be removed. Figures 1c,d show the normalized data and are much more useful.

A: We have removed Figure 1a, b.

Q8. Sixth, Tables 1 and 2 are confusing and possibly contain errors, as discussed below.

A: We have revised them according to your suggestions.

In Table 1, the excess of $^{13}C$ (not total $^{13}C$) in 100 g bulk soil is 0, and the total $^{13}C$ in 100 g bulk soil was 19.4 $\pm$0.56 mg. In Table 2, we have added the error bars.

Q9. Finally, although I agree with the authors' overall interpretation of the data, there are several sentences that are logically inconsistent throughout the manuscripts, where the statement at the beginning of the sentences does not support what follows. There is also substantial speculation and extraneous text that should be revised or removed. **These are detailed below.**

A: Thank you very much for your valuable detailed comments. We have revised them one by one.

Q10. 32-36: This statement is not logically consistent. An increase in $^{13}$C emissions does not imply lower soil organic C decomposition, nor that Rhizo-C and Micro-C soils decrease mineralization of native soil C.

A: Sorry, for not being clear in the original version. Our point is that the total C emission of Rhizo-C and Micro-C was the same as from the control soil, despite (a) the higher carbon contents in the former and (b) the fact that parts of the $CO_2$ produced in the Rhizo-C and Micro-C variants originated from the label as can be deduced from the $\delta^{13}$C ratios. Hence, the mineralization of native SOC appeared to be decreased in case of the Rhizo-C and Micro-C treatments.

We modified the sentence to: "Given the fact that about 0.3% and 0.1% of the cumulative C emission derived from the labeled Rhizo-C and Micro-C, this indicates that the soil organic C-derived emissions were lower in Rhizo-C and Micro-C treated soils than in untreated soil. This indicates that rhizodeposits and microbe-assimilated C could be used to reduce the mineralization of native soil organic carbon and to effectively improve soil C sequestration". See details at P 2, L 32-36.

Q11. 52: Heterotrophic microbes are typically much more abundant in terms of biomass than autotrophs, and would be expected to be a more important C input to SOM. This distinction is not important here.

A: Yes, heterotrophic microbes are typically much more abundant in terms of biomass than autotrophs, but particularly in paddy soils autotrophic soil microbes assimilate $CO_2$ and contribute to soil C accumulation. As this has been also addressed in our study, we think that it is valid to mention in it in the introduction.

Q12. 58-59: But you just mentioned the importance of microbes. . . green manure and manure are also often used in paddy systems.

A: We revised the statement as "The aboveground biomass and root systems of rice plants represent one of the most important inputs of available organic C to paddy SOC".

Q13. 75-77: This may be statistically significant but autotrophic microbial C fixation is equivalent to a rounding error in the total C budget of these systems. . .

A: Yes, although the amount of autotrophic microbial fixed C is relatively small, it might be worth to investigate it, as the fate of this is different than that of e.g. particulate plant-derived organic matter (roots or shoots). Microbial carbon can be particularly stabilized in soil. The wording in the paper with this respect is more careful in the revised version.

Q14. 81-83: This is a false dichotomy, as plant residues decompose to yield low molecular weight substances. 88: Contradicts the above statement, where you asserted that straw leads to priming.

A: Thanks for your comment. We have rewritten this part of the introduction.

Q15. 97: How do you define complexity here? It is unclear whether fresh plant tissue or microbial biomass would be more complex than the other in terms of biochemical composition. This hypothesis needs to be introduced and justified in the context of the literature.

A: Thank you very much for this important comment. You are right, the hypothesis is based on a not well defined and partly wrong prerequisite. Indeed microbial carbon is having a complex structure, and there is increasing evidence in literature that microbial residues are stable in soil. Hence, Microbe-C added to the soil might be relatively stable as well (which is actually also shown in the paper). In contrast, rhizodeposits exist of low-molecular and easily available organic substances. But as shown by Lu et. al. (2002) and Kuzyakov (2002), a large part of this carbon is uptaken by microorganisms and undergoing microbial metabolism. Hence, the carbon ends up partly as microbial carbon, and it is expected that Rhizo-C is having a similar fate as Microbe-C.

As we quoted in our reply to Q6, we have revised the hypothesis chapter.

Q16. 127: I disagree with this statement, microbes were definitely exposed to the $^{13}CO_2$ label given that root respiration would have been enriched in $^{13}C$. Even heterotrophic microbes assimilate $CO_2$ via anapleurotic fixation. This does not matter in the context of your treatments, and this text could be removed.

A: We have removed this text in the updated version.

Q17. 215: Delta notation should not be used for $^{13}C$-enriched samples because it is highly nonlinear away from 0 permil. The mixing analysis should be repeated using the atom percent data.

A: In the present version, the mixing analysis is based on atom percent data.

Q18. 235: This contradicts what was stated in the abstract with respect to trends in $^{13}C$ in the Rhizo C and Micro C treatments.

A: Thanks for your comment. Since we stated "but with lower percentages", this means less labeled substrate was mineralized. Therefore, it does not contradict the abstract.

Q19. 232-237: Because there was a different mass of $^{13}C$ label added to each treatment I think that Figure 1 a, b is misleading. Figure 1 c, d normalize the $^{13}CO_2$ fluxes to the amount of label added, thus the treatments can be readily compared. I recommend removing Fig. 1a, b and the associated text in the Results.

A: We have removed Fig. 1a, b and the associated text in the Results.

Q20. 250-255: Standard errors associated with these percentages are needed. 259-260: Standard errors needed

A: We have added the standard errors.

Q21. 273-274: Isn't it trivial that the cumulative $^{13}CO_2$ respired increased over the experiment? Discussing rates of change would be more informative.

A: Yes, it is certainly trivial that a substrate is mineralized with time. We deleted this phrase.

Q22. 278: Do you mean "no" effect?

A: Yes.

Q23. 304-310: This claim cannot be supported by the present data, and should be couched as speculation or removed.

A: We added the term "presumable" to this sentence, emphasizing that this is an assumption. However, there is a wealth of literature showing that such substances can get well sorbed to minerals and thereby stabilized. Some of the references are cited in the text.

Q24. 319-321: Unsupported speculation

A: Thanks also for this comment. However, here we kindly disagree. If an organic substrate consisting of a mixture of distinct organic compounds is decomposed, the more stable compounds get selectively enriched during decomposition. And it is also logically, and shown by e.g. Lu et al. (2003) and Brant et al. (2006) that a less available carbon source is used in a more conservative way, i.e. meaning that relatively less organic carbon is respired.

Q25. 322-323: But you saw PE decrease over time, right?

A: Yes, the PE decreased over time but it was still positive PE.

Q26. 328: But in natural systems, Rhizo C and Micro C typically accompany root and shoot C, they are not present on their own, unless roots and shoots are manually removed. One implication of your results might be that soil C would disproportionately benefit from shoot removal by farmers, is this correct?

A: Yes, we agree with your assessment, that in reality the C sources are not added separately into soil. However, in our study we intended to study the effect of the different C sources on the soil C mineralization and sequestration in order to identify

their potential role in priming. This can be only done, when their separate effect on priming is studied.

Q27. 330-332: Better support for this claim would come from the isotope mixing model.

**A:** We agree with your assessment, however, in our experiment we can't measure the original $^{13}C$ abundance and amount of rice rhizodeposits C and soil microbial assimilated C, because the rhizodeposits C and soil microbial assimilated C were bound to soil mineral or mixed with unlabeled SOC during the labelling period. Hence, we could not partition the amount of $CO_2$-C derived from native SOC and from rhizodeposited C and soil microbial assimilated C bound to soil mineral or mixed with unlabeled SOC during the labelling period. So we couldn't calculate their priming effect. But we could infer from the data (see also our response on Q 10) that there was a negative effect of both primers on the mineralization of native SOC.

Q28. 333: I assume you mean $^{13}C$ of $CO_2$? Need to specify here and elsewhere. , it was $^{13}C$ of $CO_2$.

**A:** Thanks, we have revised it.

Q29. 333: Unnecessary to include "rice-growing season" given that this is not a field study.

**A:** We have deleted it.

Q30. 333-336: This conclusion does not follow from the premise. This sentence is confusing and not logically consistent.

**A:** Thanks, we have revised it as follows (P 14, L 333-344): "Hence, it seems that Rhizo-C and Micro-C protects native SOC, increase the organic carbon storage of paddy soil (Ge et al., 2012; Li and Yagi, 2004; Gunina et al., 2015). Besides the lower contents of Rhizo-C and Micro-C as compared to Shoot-C and Root-C, this observation is possibly due to the different behaviour of primers in soil. Roots and shoots enter the soil as particulate and unprotected organic matter, which is to a large part well available for microorganisms; i.e. 31.9% and 45.4% where mineralized within the 300 days of incubation (Fig. S1). Rhizodeposits consist mostly of low molecular sugars and acids that are highly bioavailable (Lu et al., 2002). The relatively long MRTs of Rhizo-C (Table 2) suggests a stabilization process of this carbon, either by sorption or by microbial metabolism and recycling during the incubation (Lu et al., 2002; Gunina et al., 2014; Schurig et al., 2013). Also Micro-C hat a long MRT in the incubation (Table 2), which fits well to the observation that microbial residues are accumulating in soil (Schurig et al., 2013)"

Q31. 336-340: That is one hypothesis; another would be that these tissues are selectively stabilized due to interactions with minerals or aggregate formation. This uncertainty should be acknowledged.

**A:** Yes, you are definitively right. In the right version we consider stabilization via interactions with minerals in the introduction (in order to prepare the hypotheses) as well as in the discussion.

Q32. 346: You stated before that the PE for root treated soils was insignificant. Need to be consistent in the text, is it significant or not? If not, PE is not positive.

**A:** At early stages, there was also a positive priming in case of roots, while after loss of easily available parts of the Root-C the decomposition rates decreased and as a consequence priming was less, and considering the whole period of incubation it was

not significant. We have revised this sentence as follows: "By the end of 300-d incubation, both Shoot-C treated soils exhibited higher total mineralization and positive PEs, while Root-C failed to exhibit a significant priming effect".

Q33. 350: Should mention as a caveat that different amounts of C were added in each treatment, and it is uncertain whether this contributed to differences in the results. Table 1: The third row is unclear, why does bulk soil have 0 mg total $^{13}$C, when it is 1.08 atom percent $^{13}$C? You need to clarify or account for natural abundance $^{13}$C. Table 2: "Size" of the pools is unclear here, is this the proportion of $^{13}$C that was respired over the experiment?

**A:** Yes, you are right. And, as mentioned in our reply on Q4, now we consider this fact in the discussion.

In Table 1, the excess of $^{13}$C (not total $^{13}$C) in 100 g bulk soil was 0, and the total $^{13}$C in 100 g bulk soil was 19.4 ± 0.56 mg. And total $^{13}$C of the four photosynthesized C substrates input to 100 g bulk soil after 18 d of $^{13}$C-labelling were 11.43 ± 0.52, 5.75 ± 0.41, 1.61 ± 0.06 and 0.49 ± 0.05 mg, respectively.

In Table 2, the "Size" describes the proportion of bioavailable labelled-substrate mineralized relative to initial $^{13}$C amount.

---

## Author Comment (AC4) · 12 Jul 2016

Dear Anonymous Referee #3, thank you very much for your valuable comments. We would like to answer your concerned points one by one (Q, plain, and A, blue font).

Q1. This paper presents an incubation experiment using rice tissues and soils labeled with $^{13}$C. Labeled shoots and roots were directly added to soil. Rhizodeposits were added by shaking soil from roots, and microbe-fixed C was added by using soils that were sunlit and treated with $^{13}$C but had no plants.

This study addressed important issues related to priming effects in rice agriculture, and is appropriate for this journal. The results have important implications for C emissions, soil carbon storage, and potential strategies for mitigating greenhouse gas emissions from agriculture. The isotope labeling procedure and the incubation were sound.

A: Thank you for the positive comment.

**However, there were some issues with the analysis and interpretation that should be addressed.**

Q2. In general, the root and shoot amendments seem sound. However, the amount of labelled carbon added in the Rhizo-C and Micro-C treatments was much smaller than in the root and shoot treatments, and it's not clear whether the different treatments can be directly compared with each other. Judging from Table 2, the amount of $^{13}$C added to the soil in the Rhizo-C and Micro-C treatments was very small relative to the ambient $^{13}$C content of the soil, and I am not convinced that the emissions from these small additions were enough to be detectable in this experiment. It's difficult to tell how much labeled substrate was actually added in those two treatments, and in my opinion it calls the interpretation of results related to those additions into question. I think there should be more discussion of why the $^{13}$C emissions from these treatments can be interpreted as resulting from the amendments rather than just mineralization of ambient $^{13}$C that was already present in the soils.

**A:** As these [13]C-labeled microbial biomass or rhizosphere exudates are attached to soil minerals, it is not possible to obtain them without an associated soil matrix, unless obtained in an artificial environment. Hence, it would not make much sense to produce these substances in vitro. Of course, our approach leads to the complication that we obtained the rhizodeposits and microbial assimilated C embedded in a matrix of mineral soil with native SOM. But we still feel that our approach is appropriate to comparatively investigate the priming effects of particulate plant-derived materials and plant and microbial-derived substances sorbed or being attached to minerals within one experiment.

The total C contents of the soils containing rhizodeposited C (1.89%) and microbial assimilated C (1.90%) were larger than that in bulk soil (1.81%), though we have to admit that this difference is not statistically. But despite there was no significant difference in the total C emission between the treatments of Rhizo-C, Micro-C and CK, there was a considerable amount of label-derived $^{13}CO_2$ emitted during incubation. This indicates that the rhizodeposits and microbial assimilated C contributed to total $CO_2$ emission and, hence, the mineralization of native SOC appeared to be smaller at addition of these two primers. We do think that our approach does make sense, as the allocation pathway of rhizodeposition and autotrophic synthesis of microbial biomass is to get instantly associated with minerals.

Q3. There are issues with the equations. Most of them have typographical errors or confusing notation.

A: Thanks, we have revised all the equations. See details in L 194-223, P 9-10.

Q4. The calculation of priming effects is problematic. They are defined using cumulative emissions. However, they are then interpreted as changes over time with statements like "a positive priming effect was observed until the end of the incubation."

If calculated using cumulative emissions, any short-term priming effect would appear to last for the entire experiment, because the additional emissions at the beginning would not be cancelled out by any negative emissions later in the experiment (unless there are negative priming effects later on). Cumulative emissions could be used to calculate a total priming effect over the entire experiment in terms of extra carbon lost from SOC, but a time series of fractional priming effects like the results presented here would make much more sense if it were calculated using emission rates rather than cumulative emissions

A: We have recalculated the PE.

Q4. Line 32-36: I don't follow the logic of this statement. According to Fig. 1, emission rates from Rhizo-C and Micro-C were decreasing over most of the incubation. Cumulative $^{13}C$ emission increased over the experiment, of course, but this only means that emission rates were greater than zero.

A: Thanks. As reviewer 2 raises the same issue in his query 10, we kindly refer to our response to Q10 of reviewer 2. See details at P 2, L 32-36.

Q6. Line 75-77: These are tiny fractions. Are they really detectable in this kind of experiment? It's a factor of 100 less than rhizo-deposits.

A: Yes, we detected that phototrophic soil microbes assimilate $CO_2$ using $^{14}CO_2$ labeling method in upland soil and paddy soil (Ge et. al., 2013).

Ge, T. D., Wu, X. H., Chen, X. J., Yuan, H. Z., Zou, Z., Li, B. Z., Zhou, P., Liu, S. L., Tong, C. L., Brookes, P., Wu, J. S.: Microbial phototrophic fixation of atmospheric CO2 in China subtropical upland and paddy soils, Geochim. Cosmochim. Acta., 113, 70-78, 2013.

Q7. Line 79-81: There is a balance between microbial decomposition and mineral sorption of these substrates, and there's a lot of uncertainty about how much is respired vs sorbed over various time scales. This balance probably depends on soil physical and chemical factors, and might be different in frequently flooded soils.

A: Yes, this is certainly true. Of course, as a result of decomposition, water soluble intermediate products of decomposition are produced that can sorb on minerals. We modified the respective sentence. However, the difference between the four substrates tested is that Shoot-C and Root-C are first particulate, and just after formation of water soluble substances or resynthesis in microbial biomass, this C will get attached to minerals. In contrast Rhizo-C can be immediately bound to minerals (if not mineralized) and Micro-C is attached to minerals from the beginning. This difference is having an important impact on the mineralization rate of the substrates, as is shown by the cumulative mineralization.

Q8. Line 97-98: There aren't really any measurements of the "complexity of substrate composition" (which isn't clearly defined either)

A: Thank you very much for this important comment. As reviewer 2 raises the same issue in his query 6, we kindly refer to our response to Q6 of reviewer 2. The hypothesis chapter is rewritten at P.5-6, L.100-105.

Q9. Line 98-99: This sentence isn't very clear. Is "their relatively higher quantity and stability in soil" referring to plant residues or rhizodeposits and microbe-assimilated C? Shouldn't substrates with higher stability in soil cause weaker priming effects, because they are more resistant to decomposition?

A: Yes, the hypothesis chapter was not clear. We hope that this is clear in the revised version, which is cited in our reply to Q6 of reviewer 2.
We also expected that substrates with higher stability in soil, either due to their

inherently higher stability or due their stabilization by e.g. sorption to minerals, cause weaker priming effects because they are more resistant to decomposition. This is actually also one of our major results.

Q10. Line 144-145: The procedures for collecting rhizodeposits and microbe-assimilated $^{13}C$ sound like they include a little bit of labeled material mixed with a lot of soil, which means these additions were quite different from the plant tissue amendments, which were pure labeled tissue. This raises questions about whether the rhizodeposite and microbe-assimilated C additions are directly comparable to the root and biomass additions.

Line 164-167: The amount of carbon in these two treatments is not well known, and likely very different from the other two treatments, making direct comparison tricky.

A: Thanks for raising these two critical points. But as these $^{13}C$-labeled microbial biomass or rhizosphere exudates are attached to soil minerals, it is not possible to obtain them without an associated soil matrix, unless obtained in an artificial environment. Of course, our approach leads to the complication that we obtained the rhizodeposits and microbial assimilated C embedded in a matrix of mineral soil with native SOM. However, we do think that our approach does make sense, as the allocation pathway of rhizodeposition and autotrophic synthesis of microbial biomass is to get instantly associated with minerals. Hence, we were feeling worth to test the priming capabilities of rhizodeposition and microbial biomass together with roots and shoots within the same experiment, and our approach is appropriate to comparatively investigate the priming effects of particulate plant-derived materials and plant- and microbial-derived substances sorbed or being attached to soil minerals within one experiment.

Q11. Equation 2: I'm not an expert on isotope labeling math, but this equation looks a little strange. What is ($\delta^{13}C+100$) doing? I think it should be ($\delta^{13}C + 1000$), which

equals RS/RPDE. Either way, it seems needlessly confusing to convert RS into per-mil units and then convert that into atomic percent, instead of just using RS/(RS+1), which as far as I can tell is mathematically equivalent. Equation 3: The notation of this equation (with all the brackets and commas) is confusing. It would be easier to read with some different notation (subscripts or something).

Line 207: This is labeled equation 2 but should be equation 4. It also doesn't make sense relative to the description on lines 208-210. If y is a percentage of $^{13}C$ emission, then all of the terms in the equation should be percentages, while in fact they are pools. If $y_0$ is the pool of labeled C remaining in the soil, then it should be decreasing with time. The description of a is basically the same as $y_0$. This equation would make more sense (relative to the description of the terms) if it were $y_0 = a(1 - e^{-bx})$.

Equation 5: Should the denominator have $\delta^{13}C_{shoot}$ and $\delta^{13}C_{soil}$ rather than $\delta^{13}CO_{2shoot}$ and $\delta^{13}CO_{2soil}$? $\delta^{13}CO_{2soil}$ in the equation doesn't seem to be a thing that was actually measured. Equation 6: Priming effects are defined here as the difference in total C emissions between the amended experiments (Cshoot or Croot) and the control experiment (CK). This includes the C emissions from the decomposition of the added material as well as extra decomposition of native SOC. This is not how priming effects are usually defined, or discussed in the introduction. Usually, priming effects are defined as extra decomposition of just the native SOC, excluding emissions from the added material. If that's the case, then this equation should be isolating emissions derived from native SOC rather than using total emissions. Also, I think it would make more sense to compare emission rates rather than cumulative emissions in this ratio. If the priming effect occurs as a short pulse effect, then using this equation will exaggerate how long the priming effects last, because the increase in cumulative emissions will slowly decline as it's divided by increasing total emissions, even after increases in emissions due to priming have ceased.

A: We apologize for the mistakes in the equations, we have revised them in the revision as follows:

"The $\delta^{13}C$ values of plant residues, rhizodeposits, microbe-assimilated C, soils, $CO_2$, and $CH_4$ were    converted in $\delta$ (‰) relative to the Pee Dee Belemnite (PDB, 0.0111802) standard and further expressed in atom% as following

$$\text{atom\%} = \frac{100 * 0.0111802 * (\frac{\delta}{1000} + 1)}{1 + 0.0111802 * (\frac{\delta}{1000} + 1)} \qquad (1)$$

and the incorporation of $^{13}C$ ($^{13}C$ excess) in plant residues, rhizodeposits, microbe-assimilated C, bulk soils, $CO_2$, and $CH_4$ was calculated as follows:

$$\text{excess } ^{13}C_{sample} = [(\text{atom\%}^{13}C)_L - (\text{atom\%}^{13}C)_{UL}] \times C_{sample} /100 \qquad (2)$$

Where $(\text{atom\% } ^{13}C)_L$ and    $(\text{atom\% } ^{13}C)_{UL}$ are the $atom\ ^{13}C$ in labelled and unlabelled samples, respectively, and $C_{sample}$ are the C contents of each sample.

The $^{13}CO_2$ and $^{13}CH_4$ efflux (%) were calculated as the increases in excess of $^{13}C$-$CO_2$ and $^{13}C$-$CH_4$ within each sampling interval,, respectively, as percentages of the $^{13}C$ input. The mineralization percentage of the input $^{13}C$ was calculated as the sum of total $^{13}C$ in $CO_2$ and $CH_4$, at each sampling day, relative to the initially added total $^{13}C$.

The kinetics of the mineralization were described by fitting a first order single exponential function:

$$y = a\ (1 - e^{-bx}) \qquad (3)$$

where $a$ describes the amount of bioavailable labelled-substrate pool; $b$ is the mineralization rate of substrate; and $x$ is time (d). Obtained parameters were used to calculate the mean residence time as $1/b$ and half-life as $\ln (2)/b$.

The end-member mixing model was used to calculate the fractions of SOC- ($C_{SOC}$) and plant residue-derived C ($C_{shoot}$ and $C_{root}$), as described by Phillips et al. (2005) and Wild et al. (2014). This model allows the combination of mass spectrometric and efflux measurements. The shoot-derived $^{13}CO_2$ emission ($^{13}CO_{2shoot-derived}$) was calculated as follows:

$$^{13}CO_{2shoot-derived} = \frac{atom\%\ CO_{2shoot} - atom\%\ CO_{2CK}}{atom\%\ C_{shoot} - atom\%\ C_{soil}} \times CO_{2shoot-C} \qquad (4)$$

where $atom\%\ CO_{2shoot}$ and $atom\%CO_{2CK}$ are the atom% $^{13}C$ values of $CO_2$ derived from shoot treated soil and untreated soil (CK), respectively; $atom\%C_{shoot}$ and $atom\%C_{soil}$ are the atom% $^{13}C$ values of shoot and bulk soil respectively; and $CO_{2shoot-C}$ is the total $CO_2$ derived from shoot treated soil; and the shoot-derived $^{13}CH_4$ emission ($^{13}CH_{4shoot-derived}$) and the root-derived $^{13}CO_2$ and $^{13}CH_4$ emission ($^{13}CO_{2root-derived}$ and $^{13}CH_{4root-derived}$, respectively) were calculated similarly (Phillips et al., 2005; Ye et al., 2015).

The PE of SOM on $CO_2$ and $CH_4$ emission was calculated as follows:

$$PE_t(\%) = \frac{Gas - Gas_{CK}}{Gas_{CK}} \times 100 \qquad (5)$$

where $PE_t$ is the PE at time t (d); $Gas$ the total amount of $CO_2$ and $CH_4$ derived from native SOC mineralization in the treatment of Shoot-C and Root-C, $Gas_{CK}$ is the SOC mineralization in the CK treatment (Hu et al., 2012)."

Q12. Lines 239-241: These units don't make sense for emission rates, unless they are percent of initial $^{13}C$ lost over a specified time period (% per day or something). It's hard to interpret this without knowing what the initial $^{13}C$ was for each treatment. Those values are in Table 1, and it would help to discuss those before going into percentage losses.

A: We revised the units as $^{13}C$ efflux (% of initial $^{13}C$) d$^{-1}$. We illustrated the initial $^{13}C$ values (in Table 1) of each treatment before the $^{13}C$ loss efflux.

Q13. Lines 245-246: Based on Table 1, the initial $^{13}C$ in Rhizo-C and Micro-C treatments is nearly indistinguishable from the unlabeled bulk soil value. Are these measurements sensitive enough to determine how much of these $^{13}C$ emissions was from the labeled amendments of those treatments and how much was from the ambient $^{13}C$ content of the soil?

A: Yes, we acknowledge that the initial $^{13}C$ in rhizodeposited C and microbial assimilated C in soil was relatively small. However, we could determine the $^{13}C$ emissions from the labeled C sources by setting up a control, by which we calculated the amount of $^{13}C$ emissions derived from $^{13}C$ in rhizodeposited C or microbial assimilated C by subtracting the $^{13}C$ emissions from control.

.

Q14. Line 248-249: Why is this figure in supplemental material instead of main text?

A: Thanks. The Fig. S1 was the cumulative $^{13}C$ emissions (% initial $^{13}C$) of soils treated with different $^{13}C$-labelled carbon substrates over a 300-d incubation. The cumulative $^{13}C$ emissions was represented the sum of $^{13}CO_2$ and $^{13}CH_4$ emissions, however the $^{13}CO_2$ and $^{13}CH_4$ emissions have already shown in Fig. 2. If we added Fig. S1 to main text it could be a bit repeated.

Q15. Line 259: The methods don't describe exactly how SOC-derived C emissions were calculated. Is this the root or shoot-derived $CO_2$ emission from equation 5 subtracted from total emission?

A: We have revised the equation 5. Using this mixing-model equation we could directly calculate the C emission derived from SOC or added C source, and we could also calculate the C emission derived from SOC by subtracted C emission from added C (root or shoot) from total C emission.

Q16. Line 269: Of course the total C emissions increased. It would be impossible for them to decrease unless C emissions were negative.

A: Yes, this is a kind of trivial. In the revised version we just mention the total $CO_2$ emission.

Q17. Line 271-273: The Rhizo-C treatment included an addition of soil that was shaken off roots, so there was extra carbon. This might explain the greater total C emission.

A: Yes, surely we added some additional extra soil carbon with our method, but as the overall $CO_2$ emission did not differ, and in addition the $\delta^{13}C$ signature of the $CO_2$ shows that part of the $CO_2$ derived from the labeled Rhizo-C or Micro-C, this indicates that less indigenous organic matter was mineralized.

Q18. Line 278 should say "had no effect on the mineralization… ". Also, why do section 3.3 and the associated figure only address two of the four treatments?

A: Yes, it had no effect on the mineralization.

In our experiment we can't measure the original $^{13}C$ abundance and amount of rice rhizodeposited C and soil microbial assimilated C, because the rhizodeposits C and soil microbial assimilated C were bound to soil mineral or mixed with unlabeled SOC during the labelling period. Hence, we could not partition the amount of $CO_2$-C derived from native SOC and from rhizodeposited C and soil microbial assimilated C bound to soil mineral or mixed with unlabeled SOC during the labelling period. So we couldn't calculate their priming effect. But we could infer from the datas that there was a negative effect of both primers on the mineralization of native SOC.

Q19. Line 302-303: There is no evidence in this study to support this statement about mineral-associated organic matter. It's a possible explanation, but shouldn't be presented as a finding supported by this experiment.

Line 305-310: There also doesn't seem to be any evidence to support any of these statements. There weren't any measurements of $^{13}C$ incorporated into microbial biomass, or any of the compounds listed in line 309, so it shouldn't be stated as something found in this experiment.

A: Thanks. We revised the text in order to emphasize that this was not found in our experiment. Rather, this is information from literature that was used to explain the findings of our experiment.

Q20. Line 315: It's misleading to say that a positive PE was observed until the end of the incubation, because the PE was calculated using cumulative values. The only way the PE could stop being positive would be if there were a negative effect on total emissions later in the incubation that reduced the cumulative emissions of the amended soils. Also, it's misleading to say "with the exception of the Root-C-treated soils", since only two things are being compared. "Exception" implies that only one thing was different out of a larger group.

A: Actually, we were calculating the PE separately for all time increments during the incubation. With that we could identify that for both substrates, Shoot-C and Root-C, the PE was more pronounced at the beginning of the incubation, when more primer was available. At later stages, when the primer was having a smaller concentration and was probably microbially transformed, the stimulating effect on the native organic matter mineralization decreased. In case of Root-C the PE effect was significant at early stages of the incubation while this was not the case anymore during the later stages. We have revised the sentence as follows: "For Shoot-C, a positive PE was observed over the entire incubation period, while for Root-C this was significant only for early stages of the incubation ".

Q21. Line 322-323: Again, this statement isn't supported by PE calculated using cumulative emissions. If there is a PE observed using instantaneous emissions, it might be a more reasonable explanation.

A: We have recalculated the PE using instantaneous emissions. The PE was significant positive at initial stages of shoot- and root-C decomposition, while the PE was slowed

down at later stages, this might be the extracellular enzymes generated to degrade recalcitrant C, and promote the decomposition of SOC.

Q22. Line 327: Those differences were not statistically significant and very small, so I don't think this statement is really supported by the evidence. Certainly not enough to make such a strong statement about using them to increase SOC and mitigate global warming without stronger evidence.

A: We discussed this issue now more carefully and removed the strong statemens on mitigation of global warming. We discussed the part as follows:" Both, Rhizo-C and Micro-C augmented the C content of paddy soil (1.89 and 1.90%, respectively) over that of the untreated soil (1.81%). At the same time we found that the C emissions of Rhizo-C and Micro-C treated soils were similar to those of untreated soil. As about 0.3% and 0.1% of the substrate C, respectively, were mineralized, this suggest that rhizodeposits and microbe-assimilated C input did not stimulate native SOC mineralization but rather shows a negative priming. Hence, it seems that Rhizo-C and Micro-C protects native SOC, increase the organic carbon storage of paddy soil (Ge et al., 2012; Li and Yagi, 2004; Gunina et al., 2015)." See details in L 329-335, P 13-14.

Q23. Line 331-333: This was true for this study, but the amendments were very small. Maybe larger amendments would cause stronger effects?

A: Thanks. Yes, we agree with your assessment. The amount of $^{13}$C-rhizodeposits and microbe-assimilated C was relatively small input into soil during only 18 days continues labeling in our experiment, and might underestimate the PE.

Q24. Line 333-334: Since cumulative emissions are being shown, an increase in $^{13}$C is guaranteed as long as emissions are greater than zero, so this doesn't prove much. I don't follow the connection with suppression of native SOC mineralization. Given

that SOC contained some amount of ambient $^{13}C$, I'm not sure this result says anything about the treatment effect on SOC.

A: The total C contents of the soils containing rhizodeposited C (1.89%) and microbial assimilated C (1.90%) were larger than that in bulk soil (1.81%), though we have to admit that this difference is not statistically. But despite there was no significant difference in the total C emission between the treatments of Rhizo-C, Micro-C and CK, there was a considerable amount of label-derived $^{13}CO_2$ emitted during incubation. This indicates that the rhizodeposits and microbial assimilated C contributed to total $CO_2$ emission and, hence, the mineralization of native SOC appeared to be smaller at addition of these two primers. We do think that our approach does make sense, as the allocation pathway of rhizodeposition and autotrophic synthesis of microbial biomass is to get instantly associated with minerals.

Q25. Line 336-340: These statements are not really supported by any evidence from the experiment.

A: Thanks. We have revised these statements.

Q26. Table 1: Total $^{13}C$ should be in mg per some mass of soil, not just mg. Bulk soil total $^{13}C$ shouldn't be zero – those soils have 1% atomic $^{13}C$ and nonzero C content, so they must contain some $^{13}C$ as well. In fact, based on the numbers the amount of $^{13}C$ in the Micro-C and Rhizo-C should be very difficult to distinguish from the amount of $^{13}C$ in unlabeled soil.

A: Thanks. We are sorry for this mistake, the total $^{13}C$ in 100 g bulk soil was 19.4 ± 0.56 mg, the excess of $^{13}C$ (not total $^{13}C$) in 100 g bulk soil was 0.

Q27. Figure 1: Panels (c) and (d) show emissions in units of % of initial $^{13}C$. These units don't make sense for emission rates, unless they are percent of initial $^{13}C$ lost over a specified time period (% per day or something).

A: We have revised the units.

Q28. Figure 3: The legend is confusing because it uses "Total C in . . ." and "C derived from . . ." to refer to the same thing (i.e. cumulative C emissions). Also, "derived" is misspelled.

A: We have revised the legends.

---

## Author Comment (AC5) · 12 Jul 2016

Response to K. INUBUSHI

Dear K. INUBUSHI,

Thank you very much for your valuable comments. We would like to answer your concerned points one by one (Q, plain, and A, blue font).

Q1. This manuscript aimed to show how different C supplied from rice plant and soil microbes promoted C mineralization and interaction with native C in incubated experiment, which would be useful information to assess soil C sequestration and greenhouse gas mitigation in paddy soil, because the knowledge of C dynamics in waterlogged paddy soil is still limited to compare with aerobic upland soil (unlike L93).

A: Thank you very much for the kind comments. We have revised the sentence as follows: "There were only limited studies of estimating the fate of plant residues and rhizodeposits in paddy soils and, to our knowledge, there is no comparative information on (1) the decomposition of different organic C sources, such as rice shoots and roots, rhizodeposits, and microbe-assimilated C; or (2) the effects of different organic C sources on the mineralization of native SOC."

Q2. However one critical point of methodology is the way of preparing microbe-assimilated $^{13}C$ in soil without rice plant (L146); it is well-known that microbial activity is higher with rice plant than without plant, in general, so that the results of microbe-assimilated $^{13}C$ in this investigation might be underestimated.

A: Thanks. We acknowledge that microbial activity is higher with rice plant than without plant. In our study we wanted to separate the effects of different types of primers to native soil organic matter mineralization. As this is not possible with growing plants, we added the rhizodeposition separately. With this we hope to mimic the effects of the release of low-molecular substances by the plant and their microbial utilization. As another primer we simulated the microbe-assimilated $^{13}C$ fixed in soil by autotrophic organisms, which occurs in the paddy fallow season.

Q3. Also rhizodeposits (L144) were obtained by gentle sharking moist soil, which is common method for aerobic soil, but doubt for wet paddy soil, resulting again underestimate the contribution in $^{13}$C dymanics.

A: We apologized the not clear statement. The method of obtaining rhizodeposited $^{13}$C was revised as follows: "$^{13}$C-labelled rhizodeposits were obtained by gently shaking moist soil from the rice roots, and the soil adhere to root was washed by distilled water, then the soil slurries, consisting of soil and wash water, were mixed well and centrifuged at 13,000 g for 15 min. The fine roots with light density were removed together with supernatants".

To assess whether this method was effective enough to remove most of the fine living roots from soils, ten 15-g soil samples (two for each labeling event) were checked under a dissecting microscope. Very few fine roots were found in two samples and no roots were found in the other eight samples. We determined the $\delta^{13}$C values of two soil samples using the chloroform-fumigation extraction method (Wu *et al.,* 1990) before and after picking out the fine roots, and found no differences in $\delta^{13}$C values before and after picking out these roots. It was therefore considered that the sieving-centrifugation procedure effectively remove most of the fine living roots and was comparable to visual separation under a microscope. However, some very fine (micro) root materials may have remained in the soil samples, which may have caused minor overestimation of plant C input.

**Q4. Other minor points are as below to improve the manuscript.**

L31; Effective digits are not uniformed as 1.89 vs 1.9 and 1.8. Also L327.

A: We have revised them and used three effective digits.

L83; Yuan et al. 2014c, but only one Yuan et al. for 2014 in the list. Also L294.

A: We have deleted this reference.

L112; pH of 5.6 and a soil: water .. should be pH of 5.6 at a soil: water . . .. . .

A: We have revised this sentence.

L137; how about humidity? It is important to regulate photosynthesis/respiration.

A: During the labeling periods, the growth chambers were placed in a rice field to ensure that the environmental conditions of the labelled and control plants would be identical for labelled plants and unlabelled controls. The humidity was similar with natural condition.

L163; if the same bottle for the almost the same soil/water contents, water level should be not as in such wider range < 1-2 cm.

A: We measured the water level in bottle, it was 2–3 cm.

L167; how to adjust gas pressure during incubation especially with plant added?

A: At the initial of the incubation the gas is produced quickly, to ensure the gas pressure in bottles is similar with atmospheric pressure we sampled more frequently.

L195 C sample; unit is not clear.

A: C samples represented rice shoot, root, soil organic C, $CO_2$ and $CH_4$.

L242; linearly should be exponentially?

A: Yes, it should be exponentially and we changed it accordingly.

L249; Fig S1, missing or not available.

A: Fig S1 was provided in Supplementary Materials.

L277; different that .. should be different from ..

A: We have revised it.

L318; remove ; after Cheng et al, 2014

A: We have removed it.

Table 1; Why zero for Total $^{13}$C of Bulk soil? Negligible $^{13}$C natural abundance?

A: We are sorry for this mistake, the excess of $^{13}$C (not total $^{13}$C) of bulk soil was zero. We have added the $^{13}$C abundance of bulk soil ($\delta^{13}$C, -26.7‰) in section of "Study site and soil sampling".